# A young child formula with *Limosilactobacillus reuteri* and GOS modulates gut microbiome and enhances bone and muscle development: a randomized trial

In this randomized, double-blind controlled trial, 182 Filipino children aged 2-3 years received either an experimental young child formula (EYCF) containing a combination of *Limosilactobacillus reuteri* DSM 17938 and galacto-oligosaccharides (GOS; n = 91) or a minimally fortified milk (CM; n = 91) for 6 months. Primary outcome was tibia speed of sound and secondary outcomes were muscle strength, blood vitamin D levels, bone turnover markers, gut microbiota, fecal calcium fatty acid soaps and gastro-intestinal tolerance. Compared to CM, those in the EYCF group showed increased tibia speed of sound after 3 and 6 months. The intervention remodeled the stool microbiome composition, assessed by shotgun metagenomics, with enrichment of *L. reuteri* and higher bifidobacteria presence in the EYCF group. Increased *L. reuteri* abundance after 6 months of EYCF consumption associates with higher bone quality and muscle strength. Stool metabolomics show 45 metabolites modulated by EYCF consumption and associated to microbiome compositional changes, leading to enrichment of tryptophane and indole metabolism. In summary, consumption of EYCF containing a *L. reuteri* + GOS synbiotic improves musculoskeletal development in toddlers via modulation of microbiota composition and function. These results provide insights on gut-musculoskeletal crosstalk during early life. Clinicaltrial.gov NCT04799028.

Bone is a "dynamic" and highly specialized connective tissue, providing mechanical support for muscular activity and physical protection to tissues and internal organs, as well as a repository for the systemic mineral homeostasis[1].

Bone tissue is constantly renewed in adults as well as in children and adolescents. "Strong bones" are defined by two different aspects of bone geometry (length & width) and by the amount of mineral deposited in the bone tissue. It is interesting to note that osteoblasts first synthesize and excrete the bone matrix proteins (osteoids), which later on, at a short distance from the bone-forming cells, mineralize until about 70% of the bone, with the remaining 30% at a longer distance (6–12 months later). Therefore, bone could rapidly be under-mineralized if the mineralization rate is reduced compared to the velocity growth[2].

Most data seem to confirm that an adequate intake of calcium is important in the mineralization process to reach skeletal maturity.

✉ e-mail: marienoelle.horcajada@rdls.nestle.com

Prospective randomized clinical trials have demonstrated that calcium supplementation may increase bone mass acquisition in children, adolescence and adulthood up to the third decade of life[3,4]. When calcium supplementation ceases, the beneficial effect on bone accretion seems to disappear[5–7]. Inadequate intake of calcium may contribute to failure to develop strong bones during early life[5]. Therefore, daily recommended calcium intakes have been established by various authorities to meet the needs for growth and bone development in children[5]. In addition, nutrients (such as vitamin D, long-chain saturated fatty acids (LCSFAs)), which help calcium incorporation into bone, may also contribute to bone development[5,8–12]. Early life and toddlerhood are crucial windows for bone growth, but they are also critical for microbiome development[13]. The influence of the microbiome on bone (re)modeling has been demonstrated through different mechanisms of action, such as modulation of mineral absorption, immune cell activity, and production of microbial metabolites[14]. Given the link between the microbiome functionality and bone development, modulation of the gut microbiome by nutritional supplements has been explored to provide novel strategies to maximize bone gain during childhood through the gut-bone axis[15]. Pre-clinical growth models documented an effect of the prebiotic GOS (galacto-oligosaccharides) on mineral absorption[16], on gut hormones known to impact bone metabolism[17,18], and on bone strength and mineral density[19–21]. In clinical settings, GOS was shown to positively impact calcium absorption in adolescents[22] and post-menopausal women[23]. However, GOS has never been addressed as a specific probiotic substrate that would support bone health.

In a pre-clinical model of bone loss, a positive effect of *Limosilactobacillus reuteri* ATCC PTA 6475 on trabecular bone microarchitecture was reported[24,25]. Further, 1 year of supplementation with *L. reuteri* ATCC PTA 6475 in post-menopausal women reduced loss of tibia bone mineral density compared to placebo, suggesting that the application of select probiotic strains could be a novel approach to prevent age-related bone health and osteoporosis[26–28].

We have shown previously in vitro that the administration of a synbiotic consisting of pre-cultivated *L. reuteri* DSM 17938 and GOS stimulates microbiome metabolite production, microbial engraftment, and microbiome profiles compared to *L. reuteri* DSM 17938 alone[27]. In this study, we also confirmed the ability of specific metabolites (e.g., short-chain fatty acids (SCFAs), known to regulate bone homeostasis in vivo[28], to improve osteoblastogenesis[27]. The impact of the intervention on muscle progenitor cells remained to be evaluated.

Skeletal muscle mass and strength mainly increase from birth to adulthood[29] and are highly dependent on nutrition and exercise[30]. Muscle mass and strength increase through muscle fiber hypertrophy due to an increase of protein synthesis. The mechanism is driven by both an increase in the myonuclear protein synthesis rate and muscle progenitor cell activity, which insure myonuclear accretion. The role of muscle progenitor cells in muscle development is important, especially during the first years of life[29,31–33].

Another potential driver of muscle development is the gut microbiome, as more and more studies underline the existence of a gut-muscle axis[34], similarly to growing evidence for the gut-bone axis. It remains unknown whether administration of selected probiotic strains can impact bone and muscle development in children.

In this current study, we assessed the effect of an experimental young child formula (EYCF), containing a synbiotic blend of pre-cultivated *L. reuteri* DSM 17938 and GOS (4 g/L), in toddlers through a double-blind randomized controlled trial, compared to minimally fortified milk control milk (CM). The primary endpoint of this clinical study was tibia speed of sound, index of bone quality integrating bone density and cortical architecture, e.g., porosity and thickness[35], after 6 months of intervention. Evaluation of muscle force (handgrip) was also performed. To evaluate the potential contribution of the microbiome on bone and muscle development, the fecal microbiota composition and fecal metabolome were analyzed. Additionally, we investigated the effects of product digestion of this experimental formula, obtained via in vitro fermentation, on muscle progenitor cells activities.

## Results

We have previously shown in vitro that administration of a synbiotic consisting of *L. reuteri* and GOS improves osteoblastogenesis[27]. Prior to assessing the clinical efficacy of this synbiotic on muscle strength in toddlers, we evaluated its impact on muscle progenitor cells, key drivers of muscle development.

### Experimental formula positively impacts myogenesis in vitro

Human myoblasts exposed to the product of in vitro fermentation of the experimental formula, containing GOS + *L. reuteri*, showed a statistically significant increase in the fusion index of myotubes (+10% as compared to milk matrix alone versus a non-statistical significant increase by GOS and *L. reuteri* separately, by 4.5% and 2.4%, respectively, Fig. S1A, C) and the myotube total area (+18% as compared to milk matrix alone versus a non-significant increase by GOS and *L. reuteri* alone by 5.8% and 3.8%, respectively, Fig. S1B, C).

### Clinical trial population

A total of 273 of 278 screened toddlers were eligible for the study (Fig. 1A). Of the 273 toddlers, 182 were randomized to EYCF or a minimally fortified CM, whereas 91 remained under their habitual diet (REF). Study participants were included between June 2021 and March 2022 and completed their 6-month visit between November 2021 and August 2022. The full analysis set (FAS) included all study subjects (*n* = 138) who were randomized to the EYCF or CM. The study groups were well balanced with regard to baseline characteristics (Supplementary Data 1). All study participants were Asian. Overall, 225 (82%) subjects completed the study. In the FAS, the retention rate was 77% in both CM & EYCF groups. A total of 107 (59%) subjects completed the 6-month intervention period in the per-protocol population, of whom 51 had been randomized to EYCF and 56 to CM. Four participants dropped out in the REF group. Four participants had a major protocol deviation, defined as a compliance rate of <70%.

### Experimental young child formula increases bone and muscle strength

The mean relative change in tibia SOS at 6 months of intervention in EYCF was 58.91 m/s (95% confidence interval [CI]: 21.34, 96.48; *p* = 0.002 compared to CM) (Fig. 1C).

After 3 months of intervention, the difference was already significant, reaching 53.00 m/s (95% CI: 7.05, 98.95; *p* = 0.024 in EYCF compared to CM) (Fig. 1D). Tibia SOS was significantly greater in EYCF compared to REF after 6 months of intervention (*p* = 0.007; Supplementary Data 2, 3), while no difference was observed between the CM and REF. Bone turnover index was also significantly higher in EYCF compared to CM (+25%, *p* = 0.02) (Fig. 1E) after 6 months of intervention, while no difference was observed between CM and REF (Supplementary Data 2, 3).

Secondary outcomes, including bone growth (head circumference, tibia & radius length, Fig. 1F, G, Supplementary Data 2, 3) and body growth (height & weight for age Z score, Fig. 1H, I) were not significantly different between EYCF and CM. However, both CM and EYCF groups showed higher bone and body growth than REF after 6 months of intervention (Supplementary Data 2, 3). Finally, the muscle force of the right arm was higher in the EYCF than the CM group by 11% (95% CI: [2%, 20%]; *p* = 0.015) (Fig. 1J), after 6 months of intervention, while no difference was observed between CM and REF (Supplementary Data 2, 3).

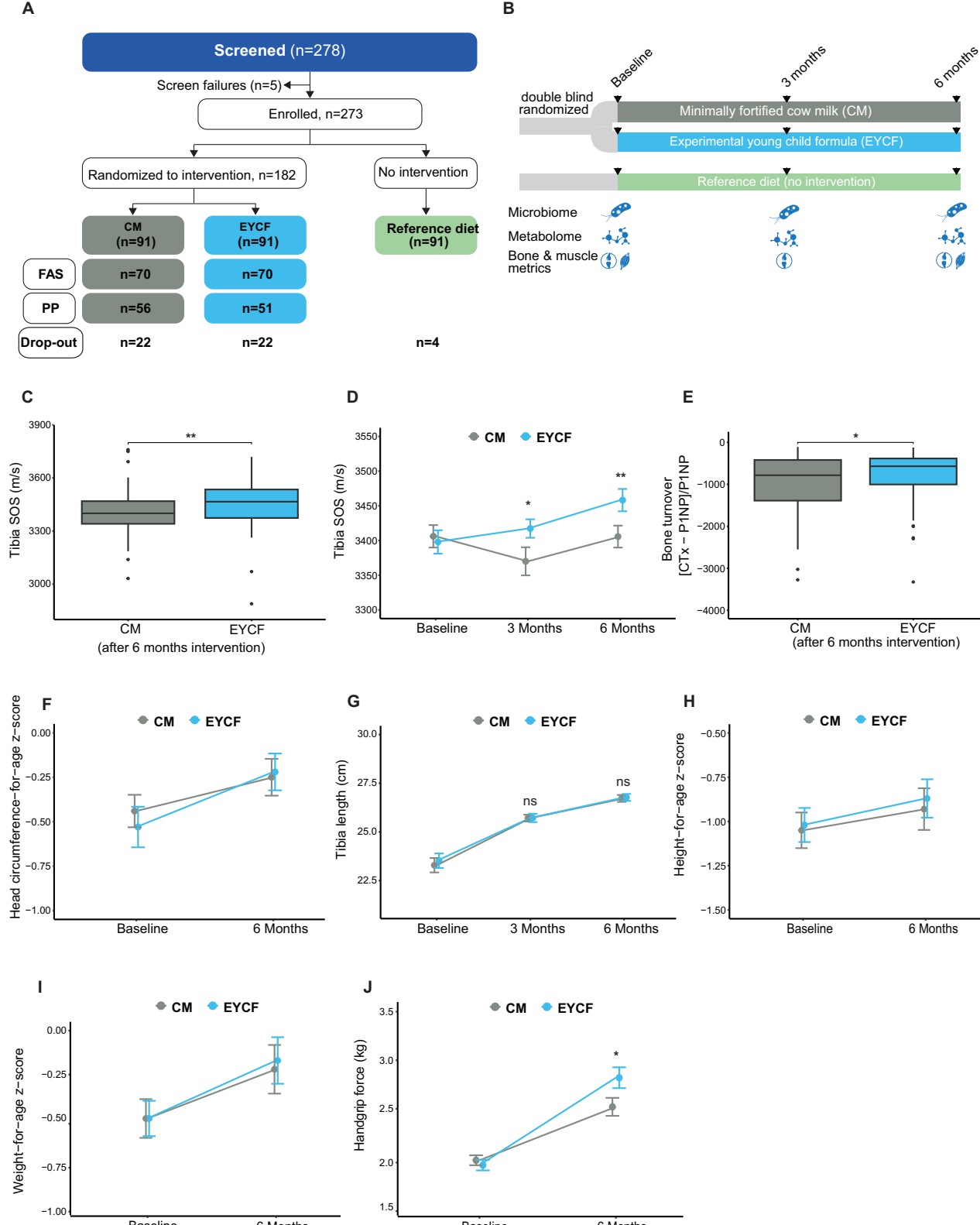

## Vitamins and minerals levels in experimental young child formula do not drive effect on bone and muscle strength

To determine whether the observed improvement of bone and muscle strength in EYCF was associated to higher vitamin levels, and/or lower intakes of sn-1,3 palmitate, we measured vitamins D, B3, B6 concentrations and excretion of fecal calcium fatty acid soaps. After 6 months of intervention, Vitamin D levels were 31.3,

36.5, and 32.7 ng/mL in the CM, EYCF, and REF groups, respectively. Vitamin D levels were 17% (95% CI: [11%, 23%]) higher in the EYCF compared to CM ($p < 0.0001$) (Fig. 2A) and 10% (95% CI: [10%, 15%]) higher compared to REF ($p < 0.001$) (Supplementary Data 4).

This effect totally abolished deficiency and reduced insufficiency of vitamin D by 52% reported at baseline (Supplemental Fig. S1D) in the

**Fig. 1 | Subjects and design of the clinical trial and effect of the experimental blend on bone and muscle clinical outcomes. A** A total of 273 toddlers were enrolled into the study; 182 toddlers were randomized to one of the feeding groups, and 91 toddlers were enrolled in parallel as a reference population. Randomized toddlers were assigned to either a minimally fortified cow milk (CM, $n = 91$) or an experimental young child formula group (EYCF, $n = 91$). Overall, 225 (82%) subjects completed the study. In the full analysis set (FAS), the retention rate was 77% in both CM & EYCF groups. Across all groups, 48 toddlers (22 each in CM and EYCF, 4 in REF) withdrew from the study with the common reasons of issues with taste ($n = 16$; 33%) and transfer to province/other country ($n = 15$; 31%). Other

reasons for withdrawal included relatives' disapproval ($n = 9$; 19%), unable to comply ($n = 4$; 8%), adverse events ($n = 2$; 4%), and preference to solid food ($n = 2$; 4%). **B** Schematic of experimental design. **C** Tibia speed of sound (SOS) measured by quantitative ultrasound after 6 months of intervention. **D** Longitudinal tibia SOS. **E** Bone turnover marker (CTx-P1NP)/P1NP. CTX: carboxy-terminal collagen crosslinks, P1NP: Procollagen type I N-terminal propeptide. Anthropometric outcomes: **F** Head circumference, **G** Tibia length, **H** Height for age, **I** Weight for age. **J** Muscle force (right arm) measured by handgrip. For all outcomes, EYCF and CM groups $n = 69$. Data are presented as mean ± SD for **C**, **E**, and mean ± SEM for **D**, **F**, **G**, **H**, **I**, **B**, *$P < 0.05$, **$P < 0.01$, compared to the CM group.

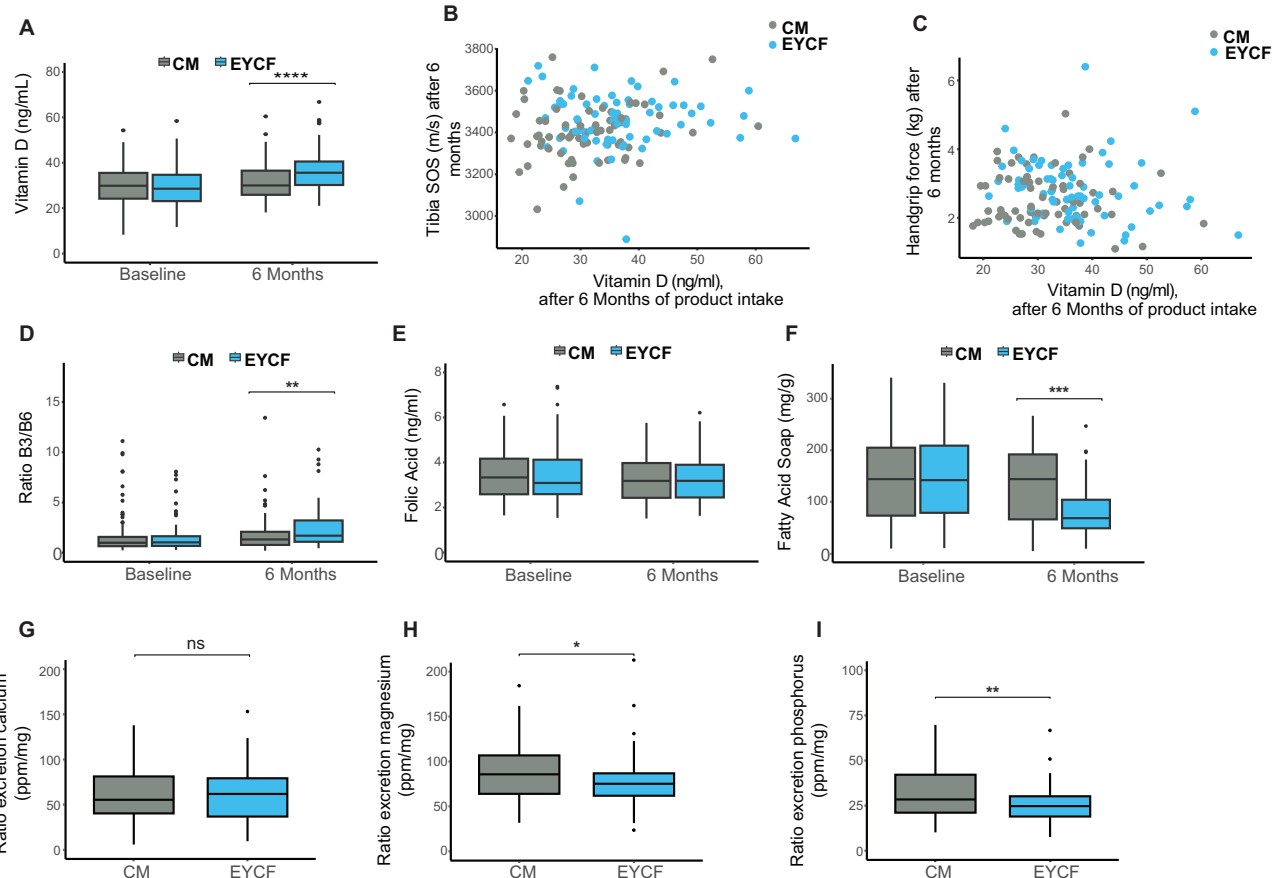

**Fig. 2 | Influence of «non synbiotic elements» in the blend on bone & muscle readouts. A** 25 hydroxy vitamin D3 levels in blood at baseline and after 6 months of intervention. **B** Lack of association between 25 hydroxy vitamin D3 and tibia SOS after 6 months of intervention. **C** Lack of association between 25 hydroxy vitamin D3 and handgrip force after 6 months of intervention. Blood vitamins measure showing the specific ratio of vitamin B3 and B6 (**D**), Folic acid (**E**) at baseline and

after 6 months of intervention. **F** Soap Fatty Acids levels in Infant Stool at baseline and after 6 months of intervention. **G–I** Respectively, calcium, magnesium, and phosphorus excretion levels in feces adjust by product intake. For all outcomes, EYCF $n = 68$ and CM $n = 69$. Data are presented as mean ± SD, *$P < 0.05$, **$P < 0.01$, ***$P < 0.001$ and ****$P < 0.0001$ compared to CM group.

EYCF. However, blood vitamin D levels were not associated to tibia SOS and muscle force (Fig. 2B, C).

A specific ratio of vitamins B3/B6, previously described to stimulate muscle regeneration[36], was increased in EYCF after 6 months of intervention (Fig. 2D). However, no association was found with tibia SOS, handgrip, radius SOS and radius length, and only a mild correlation was found with the tibia length (Supplementary Data 5). Finally, folic acid level was not different between CM and EYCF (Fig. 2E).

Total fecal fatty acids soaps levels did not differ at baseline between groups, whereas they significantly lowered after 6 months of intervention in EYCF compared to CM ($p < 0.001$, Fig. 2F). However, calcium excretion was not significantly different between groups after adjustment by intake of the product (Fig. 2G). Magnesium and phosphorus excretion appeared lower in EYCF compared to CM (Fig. 2H, I, $p < 0.01$). However, none of the calcium and magnesium minerals nor

total fatty acids soaps were associated with tibia/radius SOS, muscle force, or tibia/radius length and only phosphorus was mildly associated with handgrip (Supplementary Data 5).

## The experimental young child formula induces a bifidogenic effect

Given the important link between microbiome and musculoskeletal development in early life, we investigated whether the gut microbiome and metabolome of toddlers were impacted by the nutritional intervention.

The gut microbiome composition of all children was overall not significantly different between groups at baseline, as determined by alpha and beta diversity (Supplemental Fig. S2, Supplementary Data 6). After 6 months of intervention, toddlers consuming EYCF showed a distinct microbial community compared to baseline ($P = 0.0199$,

Supplemental Fig. S2, Supplementary Data 6). Looking at the 16 species significantly higher in EYCF compared to CM after 6 months of intervention (Fig. 3A), a notably significant increase was observed in *L. reuteri* ($Q = 6.10e–14$, effect size = 4.23), and in several bifidobacteria such as *Bifidobacterium breve* ($Q = 0.047$, effect size = 3.57), *B. bifidum* ($Q = 0.038$, effect size = 3.40), *Bifidobacterium longum* spp. ($Q = 0.021$, effect size = 2.56), *B. longum* subsp. *longum* ($Q = 0.038$, effect size = 1.85), overall suggesting a bifidogenic effect of the intervention (Fig. 3A). Other taxa significantly enriched in EYCF include 3 species belonging to the *Bacteroides* genus ($Q = 0.202$, effect size = 1.04). In CM, 21 species were significantly higher than in EYCF, including 8 species belonging to the *Streptococcus* genus ($Q = 3.84e–05$, effect size = –2.17) (Statistics for all taxonomic levels are reported in Supplementary Data 7).

## The experimental young child formula altered the microbial metabolism of amino acids

Given the changes in microbiome composition, we further investigated the impact of the intervention on its functionality by exploring the fecal metabolome (Fig. 3B).

Within it, 266 metabolites were found to significantly differ between EYCF and CM at 6-month visit, of which 69 had an unambiguous annotation (Fig. 3B). Of note, metabolites like pipecolinic acid ($Q = 2.83e–06$, effect size = 0.99), indole-3-carboxyaldehyde ($Q = 0.013$, effect size = 0.52), IPA (indole-3-propionic acid, ($Q = 0.007$, effect size = 0.61), ILA (indole-3-lactic acid, $Q = 0.04$, effect size = 0.39) and lysine ($Q = 0.001$, effect size = 0.64) were all significantly increased in EYCF compared to CM. Other metabolites strongly associated with the EYCF group included several amino acids ($Q < 0.06$, effect size 0.41–0.71), and N-acetylated amino acids ($Q < 0.09$, effect size 0.35–0.79) (Fig. 3B, Statistics for all metabolites are reported in Supplementary Data 8).

## The experimental young child formula impacts microbiome species & metabolites that are associated to clinical outcomes

We then assessed whether the metabolomic changes observed were associated with changes in microbiota composition. In this analysis, we found a significant correlation between microbiome and metabolome at all timepoints, suggesting that children with similar microbiota compositions have similar metabolomic profiles (all $p < 0.02$, correlation between 0.27 and 0.47 depending on sample subset, see Supplementary Data 9). Looking at correlations between species and metabolites significantly increased in EYCF at 6 months (Fig. 3C), we observed a majority of these metabolites correlating with the most enriched species, namely: 42 out of 45 significantly correlated with *B. breve*, 27 with *L. reuteri*, and 26 with *B. longum* subsp. *longum*—suggesting that the levels of *L. reuteri* and bifidobacteria are associated with a shift in metabolomic profiles. Of note, the abovementioned metabolites known to have an impact on bone and muscle development (i.e., pipecolinic acid, lysine, indole-3-carboxyaldehyde, IPA, and ILA) all significantly correlated positively with *B. breve, B. longum* subsp. *longum* and some to *L. reuteri* and *B. bifidum* too, although with a low effect size ($Q < 0.07$, effect size 0.18–0.41).

We further determined if the observed taxonomic and metabolic differences were associated to clinical outcomes related to bone and muscle strength. 37 species were significantly associated to at least one clinical outcome and one metabolite (Fig. 4A), including several which were significantly increased after 6 months of intervention in EYCF compared to CM. Notably, *L. reuteri* showed significant associations with 4 of the 5 measured clinical outcomes (tibia length, radius length, radius SOS, and handgrip strength, $Q < 0.07$). *B. longum* subsp. *longum* was significantly associated with tibia length and radius length ($Q < 0.004$), and nominally also with handgrip ($P = 0.02$, $Q = 0.43$). Worth noting, tibia length was the outcome associated with the highest number of species (32 out of 37, 12 of which were positively

correlated to tibia length and significantly higher in EYCF compared to CM after 6 months of intervention). All statistical associations between microbiome species and clinical outcomes can be found in the Supplementary Data 10.

We also assessed the correlation between clinical outcomes and fecal metabolites, with 15 metabolites significantly associated with at least one clinical outcome (Fig. 4A). Of interest, several metabolites that were significantly increased in the EYCF group at 6 months compared to CM were associated with at least one clinical outcome. For example, indole-3-carboxyldehyde was significantly associated with tibia length ($Q = 0.058$) and ILA with handgrip strength ($Q = 0.012$) and nominally to radius length ($P = 0.16$, $Q = 1$). Of note, 8 metabolites such as indole-3-carboxyaldehyde, tyrosine, citrulline, 3-hydroxypyridine, or 4-hydroxybenzaldehyde positively correlated with tibia SOS, although only nominally significant ($P < 0.05$, $Q <= 1$). All statistical associations between fecal metabolites and clinical outcomes can be found in the Supplementary Data 11.

To ensure metabolites from the heatmap (Fig. 4A) have a biological relevance, we tested those metabolites on osteoblast and muscle progenitor cells key markers. On top, we tested 2 key metabolites (Pipecolinic acid, indole-3-propionic acid) highly increased by EYCF (see Fig. 2B) and known from literature to impact either bone or muscle readout. All in vitro results can be found in the Supplementary Data 12. Of note, 5 metabolites, i.e., indole-3-lactic acid (+27% vs vehicle $p < 0.01$), indole-3-carboxyaldehyde (+45% vs vehicle $p < 0.01$), pyruvic acid (+40% vs vehicle $p < 0.01$), pipecolinic acid (+59% vs vehicle $p < 0.001$) and hydroxyproline (+40% vs vehicle $p < 0.01$) were able to increase osteoblastic markers (ATF-4). Four metabolites were also increasing statistically the muscle progenitor fusion index: Hydroxyphenyl lactic acid (+2.9% vs vehicle $p < 0.01$), indole-3-carboxyaldehyde (+5.2% vs vehicle $p < 0.001$), pyruvic acid (+3.2% vs vehicle $p < 0.01$), and 4-hydroxyproline (+3.2% vs vehicle $p < 0.01$) (see Fig. S3).

## *L. reuteri* abundance increases are associated with clinical outcomes

While there was an association between EYCF and tibia SOS as well as handgrip strength, this relationship was not observed in all the children in the EYCF group. To better understand this effect, we investigated whether it was driven by an increased abundance of *L. reuteri* throughout the intervention period. For this, we classified the overall study population into two categories: "increase" when *L. reuteri* abundance (derived from the associated rarefied species abundance) was increased between baseline and 6 months, and "non-increase" when *L. reuteri* did not increase. Notably, *L. reuteri* abundance increased between baseline and 6 months in half of the children in the EYCF group. Participants in which *L. reuteri* increased in abundance over time showed a significant increase of handgrip force (mean increase = 0.3 kg, Fig. 4B), tibia SOS (mean increase = 55 m/s, Fig. 4C), and vitamin D (mean increase = 3 ng/mL, Fig. 4D), with most subjects indeed belonging to the EYCF group. However, in this stratified analysis, we did not see associations with bone length or bone radius, despite their association with overall *L. reuteri* abundance. All of these trends persist when including samples from the reference group, except for handgrip force, which was no longer significant ($P < 0.1$, Supplementary Data 13).

## Consumption of EYCF is safe and well tolerated

Mean total toddler gut confort questionnaire (TGCQ) scores were low at baseline and remained low after 3 and 6 months of intervention, indicating maintenance of good GI tolerance in these healthy children. No significant difference in total TGCQ scores was observed between the EYCF and CM groups. After 3 months of intervention, mean ($\pm$SD) scores were $10.64 \pm 2.28$ in EYCF *vs.* $10.85 \pm 1.96$ in CM; $P = 0.236$, and after 6 months, scores were $10.62 \pm 1.63$ in EYCF *vs.* $10.57 \pm 1.44$ in CM;

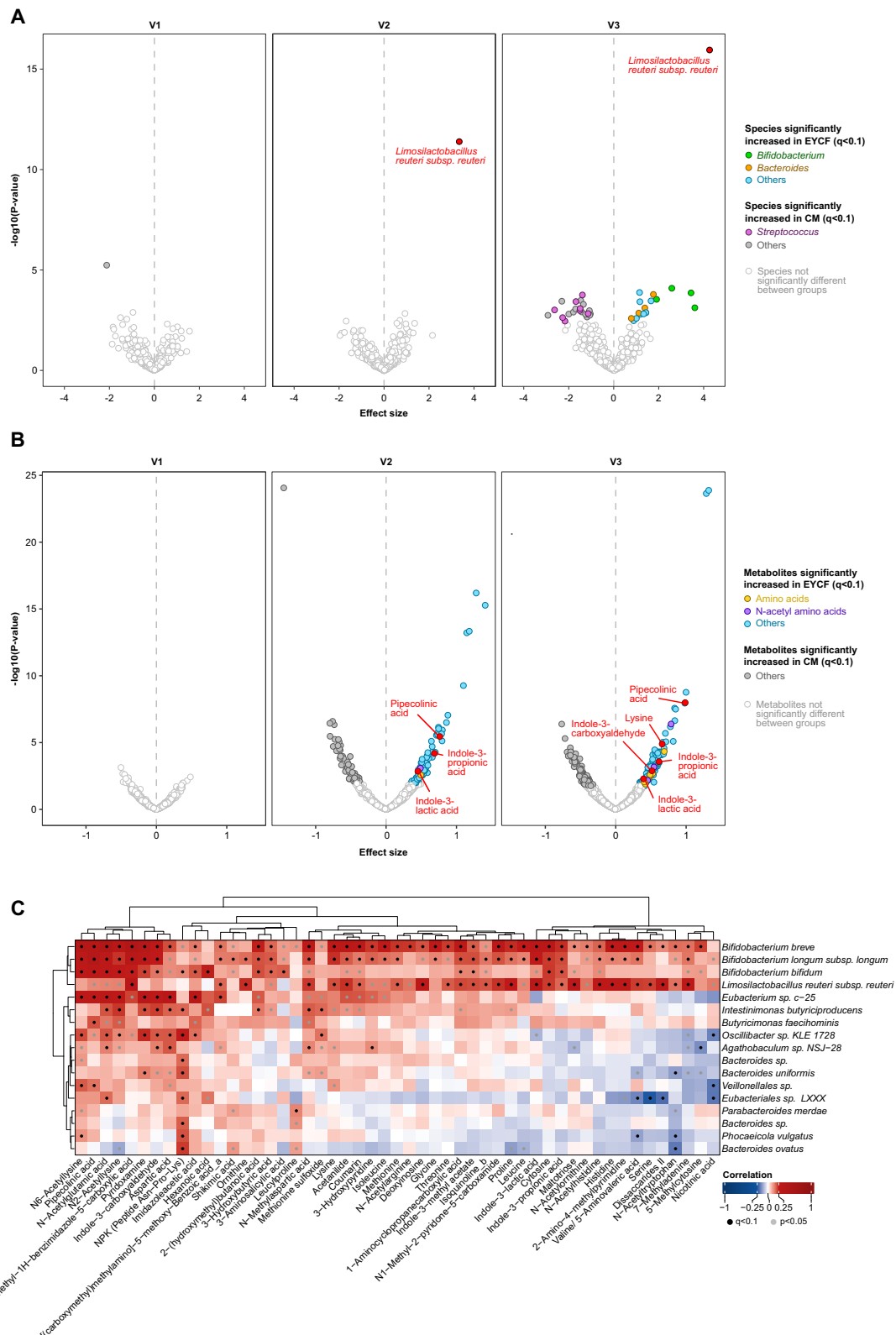

**Fig. 3 | EYCF consumption impacts both the fecal microbiome and metabolome.** Volcano plots of **A** the fecal microbiome and **B** the fecal metabolome depicting the difference between control-fed and EYCF-fed infants at each visit (421 samples in total, 208 EYCF and 213 CM). Group comparison was performed using a linear model at each visit with intervention group and sex as confounders, adding storage time as a confounder for the fecal metabolome, filtering species prevalence at 10% minimum. **C** Associations between the fecal metagenome and metabolome features significantly increased in the EYCF group at V3 using a linear model with sex and storage time as confounders.

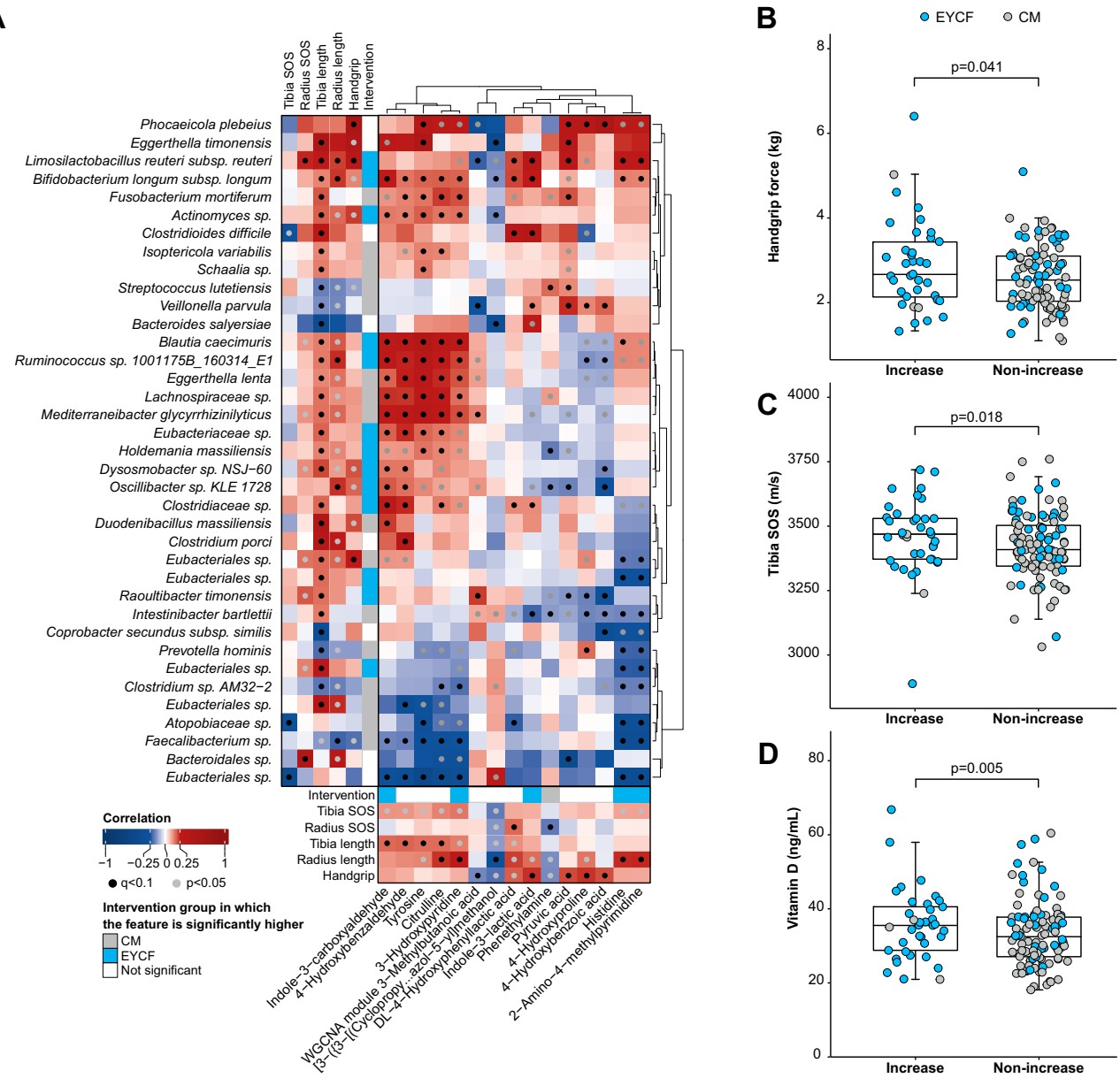

**Fig. 4 | Changes in fecal microbiome and metabolome are associated to the nutritional intervention and clinical outcomes. A** Correlations between metagenome, metabolome and clinical outcomes (see methods for more details on the statistical approach). Comparison of in handgrip force (**B**), tibia SOS (**C**), and Vitamin D (**D**) between samples showing an increase in *L. reuteri* after 6 months compared to baseline ("Increase", n = 37) and samples not showing an increase ("Non-increase", n = 101). Non-significant associations are shown in Supplementary Data 14.

*P* = 1, Supplementary Data 14, 15). The mean number of stools per day was approximately 1.3 after 3 and 6 months of intervention and not significantly different between treatment groups. Mean stool consistency scores ranged from 3.5 to 3.87 or between "mushy-soft" and "formed" at Visits 2 and 3 in both intervention groups. After 3 months of intervention, stool consistency scores were significantly lower, indicating softer stools, in EYCF *vs* CM (3.5 ± 0.62 *vs* 3.78 ± 0.39, *P* < 0.001, Supplementary Data 14, 15). There was no significant difference in mean stool consistency score after 6 months of intervention (3.72 ± 0.40 *vs* 3.87 ± 0.37, *P* = 0.079, Supplementary Data 14, 15).

The overall incidence of AEs was comparable between the groups. In the EYCF group, a total of 42 events occurred in 35 participants, while a total of 35 events occurred in 31 participants in the CM group. No AE was considered to be related to the study products, and no serious adverse event (AE) was reported in this trial.

## Discussion

Toddlerhood is a critical window for bone and muscle development, coinciding with the establishment of a more adult-type microbiome[37]. Previous reports highlight the existence of a gut-musculoskeletal axis in elderly people[15], while our study is -to the best of our knowledge- the first one exploring this axis in early life in a controlled randomized trial.

In this study, an EYCF containing a synbiotic composed of GOS and *L. reuteri* was shown to improve bone quality and muscle strength after 6 months of intake. We found that this formula increased tibia SOS compared to a minimally fortified CM at the 6-month time point (+1.5% vs CM). In addition, tibia SOS gain over 6 months was 4 times higher in the EYCF group compared to habitual diet (+1,7% vs +0.4% respectively). Handgrip strength after 6 months was increased 1.7 times compared to the control and 1.4 times compared to the habitual diet. The results of this clinical study have shown that this EYCF

containing a synbiotic of GOS and *L. reuteri* is efficient in improving bone quality (index integrating bone density and cortical architecture) and muscle strength acquisition, within normal physiological growth rates.

This is particularly true in the EYCF with previous studies in toddlers reporting increases of tibia SOS at a rate of 2.1% per year[38] and handgrip force at a rate of 30% per year[39]. In contrast, control and reference groups do not increase their tibia SOS as expected, probably due to vitamin D insufficiency, which concerns 70% of the Filipino toddler, known to be key for tibia bone quality set-up (SOS)[40]. However, the three groups have a similar rate of tibia length increase, mirroring normal growth. Furthermore, this has also been made with calcium supplementation, showing an efficacy mainly on the loaded bone[41]. Interestingly, EYCF containing 520 IU/l Vitamin D significantly increased circulating levels of vitamin D by 17%, effectively reduced insufficiency by 52% and abolished deficiency. Meanwhile, in a previous study, supplementation of 400 IU and 1000 IU, respectively, increased vitamin D levels by 8.2% and 18% in 12–30 months old toddlers[42]. The similar increase in our study to the latter higher dose suggests a potential improvement of Vitamin D absorption when co-administered with a synbiotic, which is further supported by another previous RCT combining a probiotic and vitamin D intervention[43]. However, in our study, vitamin D levels or even B vitamins blood levels did not associate to muscle strength or bone length and quality.

Fecal soap fatty acids have been shown to decrease mineral absorption in infants via the formation of fatty acid mineral soaps[44]. In our study, excretion of fatty acid mineral soaps was significantly decreased in EYCF compared to CM, but did not impact fecal calcium levels and was not associated with bone quality or muscle strength.

We also demonstrated that the supplementation with a young child formula with *L. reuteri* and GOS had a significant impact on microbiome composition both compositionally and functionally. Indeed, supplementation of EYCF with *L. reuteri* and GOS resulted in a bifidogenic effect previously reported with GOS alone[45] and *L. reuteri* alone[46], with not only *L. reuteri* being significantly increased in EYCF subjects but also several other members of the *Bifidobacterium* genus, some being reported in clinical studies to positively impact bone mass[47] and muscle function[48] in adults.

The analysis of the fecal metabolome showed that the supplementation with a young child formula with *L. reuteri* and GOS also promoted significant increases in metabolites such as pipecolinic acid, lysine, indole-3-carboxyaldehyde, known to be positively associated to bone[14] and muscle outcomes[49]. These metabolites are part of the tryptophane metabolism pathway, i.e., either direct (indoles) or indirect (pipecolinic acid, lysine) derivatives. Increased levels of indole-3-carboxyaldehyde in EYCF subjects may reflect the tryptophan metabolism of *L. reuteri*, whose genome has been described to harbor genes involved in tryptophan catabolism and which has been shown to efficiently produce indoles[50,51]. These molecules are important regulatory components of the immune system through the aryl hydrocarbon receptor signal pathway. Diet and microbial-driven changes of tryptophan metabolism via *L. reuteri* supplementation in mouse models have been shown to regulate inflammation and disease development locally in the gut as well as in the central nervous system[52]. Of note, members of bifidobacteria such as *B. breve*[53] and *B. longum* subsp. *infantis*[54] are also capable of producing beneficial indoles, including indole-3-lactic acid. Thus, a bacterial ecosystem richer in bifidobacteria may further contribute to the increased fecal indole lactate observed in EYCF. Interestingly, an increasing number of studies indicate that amino acid metabolism is crucial for bone and muscle development and homeostasis, particularly through autophagy-mediated effects[55]. Tryptophan metabolism through the kynurenine pathway is one major known factor in promoting bone-aging phenotypes, but also to stimulate bone anabolism during growth through activation of the NAD signaling pathway[56,57]. The multiple

faces of tryptophan in muscle biology also indicate the ability of tryptophan metabolites to improve muscle growth, protein synthesis, as well as antioxidant capacity, which might be partly related to myogenic regulatory factors, IGF/PIK3Ca/AKT/TOR, and Keap1/Nrf2 signaling pathways[58].

The correlation between some specific features of the microbiome and the metabolome affected by our intervention further supports the importance of using a holistic approach, including microbiome ecology and functionality, to more comprehensively assess the impact of a nutritional intervention. This analysis allowed us to show that several of the taxa and metabolites that were significantly impacted by the nutritional intervention are linked to muscle and bone metabolism. Some examples stand out, like ILA and indole-3-carboxyaldehyde, as well as precursors and metabolites related to amino acid metabolism, such as citrulline, which has been clinically shown to improve muscle performance, recovery, and protein synthesis[59,60], as well as fracture healing and mineralization in preclinical models[61,62]. In our study, we could observe some associations to health outcomes; for example, citrulline was associated with tibia SOS, and 4-hydroxyproline, a metabolite found in collagen, was positively associated with handgrip, consistent with a previous intervention with collagen peptide in sarcopenic men[63]. Also, we report for the first time, to our knowledge, two metabolite associations, 4-hydroxybenzaldehyde with bone quality and 4-hydroxybenzoic acid with muscle strength, both being present as aromatic compounds in plants.

It is interesting to note that, in our study, different metabolites associate to muscle and tibia quality. However, radius quality (a non-weight-bearing bone) and muscle strength shared associations with the same metabolites, namely positively and negatively with DL-4-hydroxyphenyllactic acid and phenethylamine, respectively. On top, we confirm that several metabolites revealed by the association with metagenomics and health outcomes have a direct biological relevance on osteoblast and/or muscle progenitor cells, like indole-3-carboxyaldehyde, pyruvic acid, and hydroxyproline. All of this highlights the importance of diverse microbes and metabolites to impact the gut-musculoskeletal axis.

In addition, we could also show a link between *L. reuteri* increase from baseline to 6 months and key clinical outcomes like handgrip strength, tibia SOS, and vitamin D, especially in EYCF. While not all participants in the EYCF group showed detectable increases in *L. reuteri*, previous studies have demonstrated that probiotic effects can occur independently of sustained colonization, namely via transient or systemic mechanisms[64]. This observation could also be explained by the limit of resolution caused by shotgun metagenomics. Further analysis using qPCR would be needed to confirm the origin of the *L. reuteri* strain, i.e., from the synbiotic intervention or from other sources, as some samples in the control group (3 out of 69 participants) also showed an increase in *L. reuteri* abundance.

Here, we provide further in vitro evidence for a synergistic effect of *L. reuteri* and GOS on muscle stem cell fusion and osteocalcin secretion from osteoblasts, two key cell processes for muscle and bone development. Even if we showed an effect of EYCF supplemented with the *L. reuteri* and GOS synbiotic on clinical outcomes and the microbiome, the synergistic effects were not assessed.

In conclusion, we have observed clinically that a young child formula supplemented with a synbiotic consisting of *L. reuteri* and GOS is safe and well tolerated, can mitigate vitamin D insufficiency, and may support healthy muscle and bone development. We further substantiated the presence of a gut-musculoskeletal axis in toddlers, as the nutritional intervention significantly affected the microbiome and the metabolome of the toddlers in the ECYF group in a correlated manner. Furthermore, several metabolites and taxa further correlated to muscle and bone outcomes. Our study additionally provides novel mechanistic insights as to how microbiome-modulating interventions can benefit bone and muscle development in early life. Whether early

gains in bone quality and muscle strength translate to persistent benefits into later childhood remains to be demonstrated.

## Methods

### Chemicals

Unless mentioned differently, all chemicals were purchased from Merck KGaA (Darmstadt, Germany). The exact references of the compounds are the following: 4-hydroxybenzaldehyde (#144088), L-citrulline (#C7629), 3-hydroxypyridine (#H57009), Indole-3-lactic acid (#I5508), L-histidine (#H6034), Indole-3-carboxyaldehyde (#129445), Indole-3-propionic acid (#57410), Pyruvic acid (#107360), Pipecolinic acid (#P2519) and 4-hydroxybenzoic acid (#H20059), Hydroxyphenyl lactic acid (Sigma, #H3253), Phenethylamine (Sigma, #128945). 4-hydroxyproline was purchased from BLD Pharmatech, #BD01052308 (PHR1939). 2-amino-4-methylpyrimidine (#A16081.09) was purchased from Thermo Fisher Scientific Inc. (Waltham, Massachusetts, USA).

### In vitro experiments with human skeletal muscle progenitors

In vitro assays were performed on primary human skeletal muscle progenitors (Lonza, Cook Myosite). Myoblasts were seeded at 90–110 cells per mm$^2$ on well plates pre-coated with 20 µg/ml human fibronectin. To evaluate the myogenesis process, cells were cultured for 3 days in growth medium (GM) (SkM-M, AMSBIO), followed by 3 days of culture in differentiation medium (DMEM/F12, Horse serum 2%, Pen/Strep 1%). Cells were exposed to the in vitro digestion of experimental blends (milk matrix +/- GOS +/- *L. reuteri*) as previously described[27] or with metabolites (Chemicals section) at 100 µM or 10 µM at each step (6 days of treatment). Experiments were performed with at least 2 donors of muscle progenitor cells, with 5–12 replicates per donor. At the end of the treatment period, cells were fixed with performaldehyde 4%, permeabilized with Triton 0.5%, blocked with 4% Bovine Serum Albumin, and stained for Troponin T assay overnight at 4 °C. The next day, the cells were washed and stained with Hoechst 33342 (Sigma) and secondary antibodies coupled with fluorescent molecules (ThermoFisher Scientific).

Image acquisition was performed with ImageXpress Micro confocal microscope (Molecular devices), and image analysis was performed with MetaXpress Software to quantify the myotube area and/or cell fusion index (myogenesis).

### In vitro experiments with osteoblast

**MC3T3-E1 subclone 4 culture and treatments conditions.** The preosteoblastic cell line MC3T3-E1 subclone 4 (CRL-2593) was purchased from ATCC (Manassas, Virginia, USA). Cells were maintained in growth medium (GM) composed of ascorbic acid-free αMEM (ThermoFisher Scientific), supplemented with 10% fetal calf serum (FCS, ThermoFisher Scientific) and 1% penicillin/streptomycin. All culture media were refreshed every 2–3 days. Cells were passaged with trypsin/EDTA solution at less than 80% confluence. To induce differentiation into osteoblasts, cells were seeded at $5 \times 10^4$ cells/cm$^2$ and grown to confluency in GM for 24 h. Then, the medium was switched to differentiation medium (DM) composed of GM supplemented with 10 mM β-glycerophosphate and treatment solutions of interest.

**RNA extraction.** MC3T3-E1 subclone 4 cells were differentiated for 7 days and were then collected for gene expression analysis. RNA was extracted using the RNeasy Plus Mini Kit (Qiagen; Hilden, Germany) with the QIAcube (Qiagen) according to the manufacturer's instructions. Briefly, cells were lysed in RLT buffer and spun in a QIAshredder column (Qiagen) before being processed by the QIAcube. RNA concentrations were measured using the Lunatic (Unchained Labs, Gent, Belgium).

**Reverse transcription and quantitative PCR (qPCR).** cDNAs were synthetized using High-Capacity cDNA Reverse Transcription Kit (Applied Biosystems; Waltham, Massachusetts, USA) according to the manufacturer's instructions. Briefly, 0.9 µg of RNA were reverse transcribed with the following program: 10 min at 25 °C, 120 min at 37 °C, and 5 min at 85 °C. cDNAs were diluted 9X with RNase-free water and used for qPCR using the PowerTrack™ SYBR Green Master Mix (Applied Biosystems; Waltham, Massachusetts, USA) according to the manufacturer's instructions. Briefly, cDNAs were diluted 5X in a solution containing the master mix and the DNA primers targeting Runt-related transcription factor 2 (*Runx2*), Activating transcription factor 4 (*Atf4*) and Catenin beta 1 (*Ctnnb1*) with following nucleotide sequences *Runx2*-f CGG ACG AGG CAA GAG TTT CA, *Runx2*-r GGG ACC GTC CAC TGT CAC TT; *Atf4*-f GGC AAG GAG GAT GCC TTT T, *Atf4*-r CGA AAC AGA GCA TCG AAG TCA; *Ctnnb1*-f TGC CTT CAG ATC TTA GCT TATGG, *Ctnnb1*-r AGA CAG CAC CTT CAG CAC. Reaction ran in a LightCycler 480 II (Roche) with the following program: 2 min at 95 °C, 40 cycles of 5 s at 95 °C and 30 s at 60 °C. The relative gene expression was evaluated via the $2^{-\Delta\Delta Ct}$ method.

**Cell proliferation.** MC3T3-E1 subclone 4 cells were seeded in GM on a 96-well plate at a density of 2'500 cells/cm$^2$. After 24 h, treatment compounds of interest were added, and the plate was transferred in the Incucyte® SX5 Live-Cell Analysis System (Sartorius Corporation, Michigan, USA) for 3 days. SYBR Green (Invitrogen, cat# S7563) was added to stain the nuclei. The nuclei count was performed with an IncuCyte mask.

### Study population

Healthy Filipino toddlers between 24 months (±1 week) to 36 months (±1 week) of age, not consuming pre- or probiotics in the past month, and without history of bone malabsorption, metabolic, congenital, or chromosomal abnormalities were eligible for this trial.

Children who fulfilled all of the following inclusion criteria were included:

1. Written informed consent has been obtained from the parent(s)/legally acceptable representative (LAR).
2. Singleton, full-term gestational birth (≥37 completed weeks of gestation), with a birth weight of ≥2.5 kg and ≤4.5 kg.
3. Child is between 24 months ±1 week to 36 months ±1 week at inclusion.
4. Child is not currently consuming nor has consumed any formulas or taking any supplements with pre- or probiotics at enrollment or in the past month.
5. Child's parent(s)/guardian is of legal age of consent, must understand the informed consent and other study documents, and is willing and able to fulfill the requirements of the study protocol.

Exclusion criteria that rendered children ineligible for inclusion:

1. Chronic infectious, metabolic, genetic illness, or other disease including any condition that impacts feeding or growth.
2. History of bone malabsorption, metabolic, congenital, or chromosomal abnormality known to affect feeding or growth.
3. Use of systemic antibiotics or anti-mycotic medication in the 4 weeks preceding enrollment.
4. Known or suspected cows' milk protein intolerance/allergy, or lactose intolerance, or severe food allergies that impact diet.
5. Current breast milk feeding in place of all other milk, and/or milk alternatives.
6. Clinical signs of severe micronutrient deficiencies.
7. Parent(s) not willing/not able to comply with the requirements of the study protocol.
8. Child's participation in another interventional clinical trial.

No changes in eligibility criteria were performed after trial commencement.

## Study design

The study is a randomized, controlled, double-blind intervention conducted in healthy male and female children, which were exposed to either minimally fortified cow milk (CM) or to EYCF. A non-randomized habitual diet reference group was also included from enrollment to 6 months (REF). The study was conducted at the Asian Hospital and Medical Center in Muntinlupa City, Philippines, from June 2021 to August 2022. The Asian Hospital and Medical Center Research Ethics Committee approved the study (REF:QF-REC-007). Written informed consent from all parents or LARs was obtained prior to screening and enrollment. The trial was registered at clinicaltrial.gov as NCT04799028.

Children completed three visits (enrollment, after 3 months, and after 6 months). Stool samples were collected at all three visits. The primary endpoint was tibia bone mass index measured by speed of sound (SOS) at 6 months of intervention. Secondary endpoints included additional bone mass index parameters, muscle force (Handgrip strength) gastrointestinal (GI) tolerance, and stooling patterns, excretion of fecal calcium, fatty acid soaps, serum vitamin D, urine and blood bone turnover markers, gut microbiota, fecal metabolism, and safety endpoints including growth, medication use, and standard AEs. A schematic of the trial design is presented in Fig. 1A, B.

## Sample size

The sample size calculation was based on the primary endpoint—tibia SoS. Assuming an expected difference of 70 m/s between the two study groups, with a standard deviation of 133 m/s[65], an overall type I error of 5 and 90% statistical power, the number of completed subjects per milk feeding group is equal to 77. Assuming a drop-out rate of approximately 15% over the 6 months, 273 toddlers (approx. 91 per feeding arm) were enrolled. The drop-out rate was to be monitored during the conduct of the trial, and the number of enrolled subjects will be adjusted to ensure 231 completed subjects (77 subjects/arm).

## Product allocation, product description, groups, and timing

Toddlers were randomly assigned to the control (CM) or experimental (EYCF) formula. Children were fed the assigned formulas for 6 months from enrollment. The nutritional composition of EYCF and CM is within regulatory limits for children's formula and fortified milks, respectively.

- CM: Powdered cow's milk, fortified with the key nutrients (vitamins A, D, E, and C), with nutritional composition all within regulatory limits and an average of nutrient levels found in commonly consumed children's milk in Philippines (Department of Science and Technology—Food and Nutrition Research Institute (DOST-FNRI). 2016. Philippine Nutrition Facts and Figures 2015: Dietary Survey).
- EYCF: Powdered cow's milk-based toddler formula containing calcium (120 mg/100 mL), Vitamin D (61IU/100 mL), GOS (4 g/L), pre-cultivated *L. reuteri* DSM 17938 (at a concentration that guarantees $10^8$ cfu/day), and a fat blend low in sn-1, −3 LCSFAs. Pre-cultivated *L. reuteri* consists is to prime the bacterial enzymatic machinery of a probiotic strain for a specific prebiotic, here GOS, by having a structurally similar substrate as a carbon source during production. We previously demonstrated that pre-cultivated *L reuteri* lead to improved ex vivo metabolic profiles, microbial community dynamics, probiotic engraftment, and ex vivo osteoblastogenesis[27].

Since EYCF is a toddler formula tailored to the needs of young children, the nutritional profile of EYCF is slightly different from that found in CM, especially pertaining to protein levels. The nutritional composition of the CM and EYCF is summarized in Supplementary Data 16.

Children randomized to CM or EYCF received the feeding orally, ad libitum, every day for 6 months from enrollment. Recommended consumption in line with dietary guidelines was at least 2 servings per day (~235 mL of reconstituted product per serving). Children in the reference group continued their habitual diet.

## Clinical readouts

**Anthropometry.** Anthropometric parameters were collected at baseline and 6 months after intervention, including weight (kg), height (cm), and head circumference (cm); BMI was calculated (kg/m²). Children were weighted without clothing or a diaper on a calibrated electronic weighing scale, and measurements were recorded to the nearest gram. Height was measured using a stadiometer to the nearest 0.1 cm. Head circumference was measured using a pediatric non-elastic tape measure to the nearest 0.1 cm. All anthropometric measures were repeated until reproduced within a pre-defined acceptable range (i.e., 10 g for weight, 0.5 cm for height and 0.2 cm for head circumference). For the baseline and 6-month timepoint, corresponding WHO child growth standard z-scores, including weight-for-age, height-for-age, head circumference-for-age, and BMI-for-age, were computed. Radius and tibia length were determined using a ruler, as previously described[66].

**Speed of sound of the tibia and radius.** SOS readings were measured using the Sunlight Omnisense® 9000S Bone Sonometer (BeamMed Ltd) as previously described[67,68] at baseline, 3 and 6 months after intervention. The Sunlight Omnisense® is a portable ultrasonometer which emits and receives pulses of ultrasound along the radius bone at a low 200-kHz frequency and measures the SOS in the radius (low-frequency velocity, VLF, m/s). This device is designed to measure the SOS of bones that are close to the surface of the skin, such as the tibia or radius. Ultrasound transmission gel (Aquasonic 100 Ultrasound Transmission Gel; Parker Laboratories Inc.) was used to obtain good acoustic contact between the probe surface and the soft tissue overlying the tibia. Two trained investigators worked on VLF assessments and performed two measurements on four same participants. Inter-observer CV was 2.07% for tibia and 4.16% for radius.

**Muscle force as measured by Handgrip digital dynamometers.** Force was measured with the Jamar hand dynamometer at baseline and 6 months after the intervention as previously described[69]. The child was seated upright on a chair in front of the instrument, which was placed on a table. The most used posture was as follows: shoulders adducted and neutrally rotated, elbow flexed at 90°, forearm in neutral, and wrist between 0 and 30° of dorsiflexion. The child grasped the handle and was allowed to become familiar with the instrument by obtaining a good grip, squeezing lightly, and watching the corresponding increase in grip strength on the digital display. When ready, the child was asked to squeeze as hard as possible for 10 s on a verbal "go" signal. Three trials for each hand were conducted, alternating hands, and always starting with the dominant hand. There was always a break of at least 1 min between the tests on the same hand. The results of each of the three tests per hand were noted in a test protocol. Some systematic verbal instruction was given, like: "I want you to hold the handle like this and squeeze as hard as you can". The examiner demonstrated and then gave the dynamometer to the subject. After the child was positioned appropriately, the examiner said, "Are you ready? Squeeze as hard as you can". As the child began to squeeze, the examiner said, "Harder!... Harder!... Relax"[70].

**Gastrointestinal tolerance and stooling patterns.** GI tolerance was assessed using the Toddler Gut Comfort Questionnaire (TGCQ) at baseline, after 3 and 6 months of intervention. It consists of 10

questions pertaining to GI-related symptoms and behaviors, including constipation, diarrhea, gassiness, abdominal pain, difficult to pass/hard stools, bloating, fussiness, and sleep problems that are recalled retrospectively over the past week from a parent's perspective. Possible scores for each individual question on the TGCQ range from 1 (Never/None/Not a problem at all) to 6 (Always/Very strong). Scores for each of the 10 questions were summed to provide a total score with a possible total score of 10–60. Stooling patterns, which includes stool frequency and consistency, were recorded via the GI symptom and Behavior Diary at baseline, after 3 and 6 months of intervention. Stool consistency was rated by parents on a 5-point stool scale: 1 = watery, 2 = runny, 3 = mushy soft, 4 = formed, and 5 = hard.

## Blood markers

Subject's blood bone turnover marker levels, including C-terminal cross-linked telopeptides of type I collagen (CTX) and procollagen type I N-propeptide (P1NP), were measured at baseline and 6 months after the intervention as described in ref. [71]. Bone turnover index, define as (CTX-P1NP)/P1NP, was calculated as described in ref. [71].

## Blood vitamins

Sample preparation, including calibration curves, quality controls (QC), and study samples, was automated and performed on a Microlab Star M liquid handler (Hamilton, Reno, NV, USA). Briefly, samples were thawed at room temperature, vortexed, and transferred to polypropylene plates containing internal standards (IS) and Ascorbic Acid. Samples were precipitated using 7.5% trichloroacetic acid (TCA), vortexed during 10 min and finally centrifuged at 2500 rpm for 10 min. Supernatant was transferred and filtered onto an AcroPrep Advance 96 filter plate with a 0.2 μm membrane (Pall, Port Washington, NY, USA) prior to LC-MSMS analysis.

Water-soluble B vitamins (B3, B6, and B9) analyses in CM and EYCF groups were performed on an Acquity I-class UPLC system (Waters, Milford, MA, USA) composed of a binary solvent pump, a sample manager with a fixed-loop injection system (SM-FL) set at 6 °C, and a column oven equipped with an active preheater set at 40 °C. Separations were performed on an ACE Excel, 1.7 μm, C18-AR 100 ×3.0 mm column (ACE, UK) in gradient mode using solutions containing 1% Acetic acid, 0.4% Formic acid, and 0.2% Heptafluorobutyric acid (HFBA) in Milli-Q water (Merck®, DE), and Methanol as mobile phases. A constant flow rate of 450 μL/min was used, and a volume of 2 μL was systematically injected. The UPLC system was hyphenated to a Xevo TQ-XS triple quadrupole mass spectrometer (Waters, Milford, MA, USA) equipped with an Electrospray Ionization (ESI) source. Argon was used as a collision gas, and multiple reaction monitoring (MRM) transitions were experimentally determined for each compound. Data were acquired using MassLynx software (Waters, Wilmslow, UK), and chromatographic peaks were integrated with TargetLynx (Waters, Wilmslow, UK). A calibration curve is built using the response (ratio of analyte area and IS area) and the theoretical concentration of each calibration point, then weighted. QCs and samples are quantified using the calibration curve and reported in ng/ml.

For vitamin D (25-hydroxycholecalciferol) levels, blood samples were centrifuged and sent on ice to Asian Hospital and Medical Center in Muntinlupa City. Concentrations of serum 25(OH)D were measured using the DiaSorin radioimmunoassay (RIA) method. The intra-assay coefficient of variation (CV) was less than 2%. All laboratory technicians were blinded to the case status. Deficiency was defined as levels below 20 ng/mL and insufficiency as levels between 20 and 30 ng/mL.

## Measurement of fatty acid and mineral excretion in feces

A 27 g stool sample was collected by the parents at home and stored in the freezer until picked up by study staff. The samples were stored locally in a −80 °C freezer at the study site in the Philippines, and shipped on dry ice to Eurofins (Madison, WI, USA) for analysis of total

fatty acid soaps and mineral content (total calcium, magnesium, and phosphorus), in CM and EYCF groups. The dried stool sample was extracted, and neutral lipids, including non-soap free fatty acids, were obtained by solvent reflux. The remaining sample was treated by acetic acid to release the fatty acid soaps, which were obtained by a second solvent reflux step, as described previously[72]. Total fatty acid soaps were calculated based on the sum of all measured individual fatty acid soaps and were normalized to grams of dry stool weight in the acid form and expressed as mg/g. The limit of quantification for major fatty acids is a standard 0.05 mg/g based on a 0.5 g sample weight. Mineral analyses were performed by ICP.

## Statistical analysis on primary and secondary outcomes

The primary outcome was tibia SOS measurements after 6 months of product intake. The estimated effect for the primary endpoint was obtained using a mixed model for repeated measurements (MMRM) for the tibia SOS measurements at 3 months and 6 months. The fixed covariates of the MMRM were: randomized treatment (EYCF or CM), visit (3 months or 6 months), visit in interaction with treatment, baseline tibia SOS measurement (at enrollment), and sex. Participant identification number was used as a random effect to account for the intra-participant correlation. No imputation methods were used, assuming that the data were missing at random, and that the resulting estimator of the mixed linear model is unbiased under this assumption. Significance of the estimated effect for the primary endpoint was tested at 5% level, and no multiplicity adjustment was needed.

All conclusions for the primary outcome were based on the intention-to-treat analysis population (ITT), which quantified the average treatment effect in all randomized participants regardless of adherence to treatment. A supportive analysis for the primary outcome was carried out on the per-protocol set (PP), consisting of all children with at least 70% compliance for the product intake, no consumption of other fortified milk, and no intake of vitamin supplements. No difference on the primary outcome between ITT and PP analysis was observed.

Continuous secondary endpoints were analyzed using MMRM when repeated measurements were performed at both 3-month and 6-month visits, with the same covariates as for the primary endpoint. For continuous secondary outcomes measured only at baseline and at the end of the study (6-month visit), an analysis of covariance model was used to estimate the treatment effect. The factors of the analysis of covariance model were randomized treatment, sex, and baseline endpoint. When necessary, a natural logarithm transformation was performed to meet the assumptions of homoskedasticity of residuals. Estimated effects of secondary outcomes were tested at 5% without adjustment for multiplicity. No imputation for missing data was performed on secondary endpoints.

Associations between different clinical outcomes were performed using both Pearson and Spearman correlation coefficients; the significance of the correlation was tested at 5% level without adjustment for multiplicity.

Subjects in the formula group were randomized to receive the EYCF or the minimally fortified milk. Subjects belonging to the reference group (REF) were not randomized; they were recruited in the trial and continued their habitual diet, which may or may not consist of cow's milk-based products.

In this situation, confounding can occur if some covariates are related to both the treatment assignment (exposed to either EYCF or CM) and the outcome. In the presence of confounding, propensity score methods were used to remove the effects of confounding when estimating the effect of treatment. For this analysis, sex, gestational age (GA), delivery mode, breastfeeding history, age at enrollment (months), and BMI at baseline were used as variables to build the propensity scores.

First, a comparative model without adjustment was carried out. After which, a model including the computed propensity scores as a covariate was carried out. Then, a model including the computed propensity score as a weight (inverse probability of treatment weighing method) was carried out. The three models included as covariates by default the sex, GA, weight at birth, delivery mode, breastfeeding history, age at enrollment (months), and BMI at baseline. The comparison between the first and the next two models highlighted a part of the bias introduced by the lack of randomization. The results were reported when the three models lead to similar conclusions, for a robust interpretation of the results.

## Microbiome analysis

Out of 278 screened subjects, a total of 220 participants (81%) provided stool samples at all timepoints, while 41 participants (6%) provided samples only at baseline. Samples were processed similarly to Capeding et al.[73]: DNA was extracted from -0.1 g aliquots of the fecal samples using the NucleoSpin 96 Stool (Macherey-Nagel) kit. Bead beating was done horizontally on a Vortex-Genie 2 at 2700 rpm for 5 min. A minimum of one negative control was included per batch of samples from the DNA extraction and throughout the laboratory process (including sequencing). A ZymoBIOMICS Microbial Community Standard (Zymo Research) was also included in the analysis as a positive (mock) control. Before sequencing, the quality of DNA was evaluated using agarose gel electrophoresis, and the quantity of DNA was evaluated by Qubit 2.0 fluorometer quantitation. The genomic DNA was randomly sheared into fragments of around 350 bp, then used for library construction using NEBNext Ultra Library Prep Kit for Illumina (New England Biolabs). DNA libraries were evaluated using Qubit 2.0 fluorometer quantitation and Agilent 2100 Bioanalyzer for the fragment size distribution. Quantitative real-time PCR (qPCR) was used to determine the concentration of the final library before sequencing. Libraries were then sequenced using $2 \times 150$ bp paired-end sequencing on an Illumina platform. After removing low-quality reads and reads mapping to the human genome (GRCh38), the remaining reads were aligned to the Clinical Microbiomics in-house extended infant fecal microbiome gene catalog (20,992,486 microbial genes), allowing for taxonomical profiling of a corresponding set of 1472 metagenomics species (MGS)[74]. All MGS relative abundances are shared in Supplementary Data 17.

## Metabolome analysis

Stool metabolomics were performed untargeted for semi-polar metabolites following an analytical protocol adapted to a UHPLC system (Vanquish, Thermo Fisher Scientific) coupled with a high-resolution quadrupole-orbitrap mass spectrometer (Orbitrap Exploris 240 MS, Thermo Fisher Scientific). Data were processed with Compound Discoverer (3.2, Thermo Fisher Scientific) and MS-Omics software, with compound annotations based on the MS-Omics internal spectral library, mzCloud, the Human metabolome database (HMDB, version 5.0), and the FooDB.

Short chain fatty acid analysis of Acetic acid, Formic acid, Propanoic acid, 2-Methylpropanoic acid, Butanoic acid, 3-Methylbutanoic acid, Pentanoic acid, 4-Methylpentanoic acid, Hexanoic acid, Heptanoic acid, and p-Cresol was performed in a targeted manner on a high-polarity column (ZebronTM ZB-FFAP, GC Cap. Column 30 m × 0.25 mm × 0.25 μm) installed in a GC (7890B, Agilent) coupled with a time of flight MS (Pegasus® BT, LECO). Raw GC-MS data were processed and quantified in Skyline 22.2 (Adams et al., PMID: 31984744) and the PARADISe software (v2.6) developed by MS-Omics and collaborators. Finally, clusters of co-abundant compounds were identified using the weighted correlation network analysis (WGCNA) framework as implemented in the WGCNA R package (Langfelder & Horvath, 2008), applied on the merged data set including the SCFA processed peak areas and the semi-polar processed PQN peak areas. A signed, weighted compound co-abundance correlation network using biweight midcorrelation, with <5% of the individuals regarded as outliers, was calculated across all included samples using all pairwise observations. A scale-free topology criterion (R2-cutoff for scale-free topology = 0.8) was used to choose the soft threshold, resulting in $\beta = 41$. Clusters of positively correlated compounds were identified with the dynamic hybrid tree-cutting algorithm[75], using a deepSplit of 2, a minimum cluster size of 10, and the partitioning around medoids option turned on. The profile of each metabolite module was summarized by the module eigenvector, i.e., the first principal component of the metabolite abundances across samples. Pairs of modules were subsequently merged if the correlation between the modules' eigenvectors exceeded 0.85. The resulting metabolite modules were named "MXX", where "XX" is an integer. Compounds that did not fit the clustering criteria were assigned to a leftover group named "M0". Complete metabolite data is available upon request.

## Integration of microbiome, metabolome, and clinical outcomes

Microbial abundances for taxa present in at least 10% of samples were centered log-ratio transformed after replacing all zeroes with half the lowest observed non-zero abundance per taxon[76]. Semi-polar metabolite intensities were power transformed (Box-Cox). SCFA quantifications were log2 transformed. To be robust against outliers in the data, winsorization was performed, capping the transformed abundances at their 3rd and 97th percentile.

The intervention effect on the microbial abundances and the metabolite intensities was estimated with cross-sectional linear models with the covariates: intervention group (CM or EYCF) and sex.

Correlations between microbial abundances, metabolite intensities, and clinical outcomes were estimated on samples at all time points and the two intervention groups collectively using linear mixed models adjusting for sex as a fixed effect and participant as a random effect, with effect sizes reported as partial Pearson correlation coefficients (obtaining by scaling the T-values of the linear model).

All models including metabolite intensities have the additional fixed effect covariate: storage time of the metabolome sample. All models including microbial abundances used a compositional bias correction based on LinDA[77].

P-values were adjusted for multiple testing by the FDR method[78]. These adjusted P-values are referred to as Q-values. Adjustments were performed separately for each visit, each microbial taxonomy level, and for the set of WGCNA modules, as well as for each annotation level of the metabolites (SCFA, 1, 2a, 2b, 3), where relevant.

The alpha diversity of the microbial composition was estimated with richness and Shannon index, both estimated at the gene-level and species-level, and with species-level Faith's PD. Weighted UniFrac distances were used for beta diversity analysis.

Procrustes analysis was performed on the first two principal coordinates of the weighted UniFrac distance matrix (microbial composition) and the first two dimensions of a redundancy analysis (for metabolite intensities) using protest from the vegan package with 999 permutations.

## Association of *L. reuteri* increase vs. non-increase groups with clinical outcomes

Statistical comparisons between samples displaying an increase in *L. reuteri*, or a lack thereof, between baseline and after 6 months of intervention, were run using the same approach as in the section "Statistical analysis on primary and secondary outcomes".

## Reporting summary

Further information on research design is available in the Nature Portfolio Reporting Summary linked to this article.

## Data availability

All clinical outcomes data necessary to interpret the results are included in supplementary data files or are available in a deidentified format upon request to the corresponding author (Dr. MN Horcajada, marienoelle.horcajada@rdls.nestle.com). Shotgun metagenomics data (depleted from reads representing the human genome) were deposited in the European Nucleotide Archive under BioProject: PRJEB83333. The metabolomics data have been deposited to MetaboLights repository with the study identifier MTBLS12557. For both PRJEB83333 and MTBLS12557 a mapping file summarizing ID number from subject and visit to sample has been added. No expiration date of data request is currently set once data are made available.

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

## Acknowledgements

Authors would like to thank Nestlé Research colleagues: Olivier Ciclet and Adrien Frezal for technical expertise on B vitamins analyses, Dustin Larue and Francesca Giuffrida for fatty acid soap analysis and interpretation, Héloise Denurra for metabolomics results interpretation, Ute Haeberlein Schwan for experimental product development, Flavien Bermont and Aurelie Hermant for technical support on bone biomarkers analyses, and Eugenia Migliavacca for her support on statistical analysis. We would like to thank Eline van der Beek, Irma Silva-Zolezzi, and Laurence Biehl for critical review of the manuscript. We thank Helle Pedersen and Thorsten Gravert (Cmbio) for their expert insights on metabolomics and statistical models associated, and Claire Boulangé for supporting metabolomics interpretation. Societé des Produits Nestlé SA, Nestlé Research, Lausanne, CH.

## Author contributions

N.B., Y.C., I.J., and M.N.H. designed the study. M.R.C., J.L., and L.C.P. were investigators in the study, were responsible for recruiting participants and for collecting data. N.B., T.D., M.B., J.N.F., and M.N.H. were responsible for analyses of bone, muscle outcomes, circulating levels of vitamin and fecal fatty soap and minerals. L.F.K. generated the random allocation sequence, had access to raw data and was responsible for statistical analyses. N.B., L.S., T.D., H.L.P.T., L.F.K., D.E., and M.N.H. drafted the manuscript and designed figures. N.B., D.E., L.F.K., T.D., L.S., and M.N.H. have directly accessed and verified the underlying data reported in the manuscript. J.M.M., A.G., and P.R.G. generated, analyzed, and integrated microbiome and metabolome data. MGG oversaw the microbiome analysis. M.G.G., L.S., and H.L.P.T. interpreted microbiome, metabolome, and clinical integration results. All authors had full access to all the data in the study and had final responsibility for the decision to submit for publication.

## Competing interests

Some authors (N.B., L.S., M.G.G., T.D., H.L.P.T., L.F.K., J.L., M.B., J.N.F., I.J., Y.C., D.E., and M.N.H.) are employees from Société des Produits Nestlé SA. Cmbio employees (J.M.M., A.G., and P.R.G.) contributed within a service agreement contract. M.R.C. and L.C.P. had no conflict of interest to report.

## Additional information

**Nicolas Bonnet** [1], **Maria Rosario Capeding** [2], **Léa Siegwald** [1], **Marc Garcia-Garcera** [1], **Thibaut Desgeorges** [1], **Hanne L. P. Tytgat** [1], **Laura-Florina Krattinger** [3], **Jowena Lebumfacil** [4], **Loudhie Cyd Phee** [2], **Janne Marie Moll** [5], **Alexander Gudjonsson** [5], **Paula Rodriguez-Garcia** [5], **Michael Baruchet** [1], **Jerome N. Feige** [1], **Ivana Jankovic** [6], **Yipu Chen** [6], **Delphine Egli** [6] & **Marie-Noëlle Horcajada** [1] ✉

[1]Nestlé Institute of Health Sciences, Nestlé Research, Vers-Chez-Les-Blanc & EPFL innovation Park, Lausanne, Switzerland. [2]Clinical Research Unit, Asian Foundation for Tropical Medicine Inc. (AFTMI), Muntinlupa City, Philippines. [3]Clinical Research Unit, Nestlé Research, Lausanne, Switzerland. [4]Wyeth Nutrition, Makati City, Philippines. [5]Cmbio, Copenhagen, Denmark. [6]Nestlé Product Technology Center—Nutrition, Société des Produits Nestlé S.A., Vevey, Switzerland. ✉e-mail: marienoelle.horcajada@rdls.nestle.com

