## [Transparent Peer Review file · Nature Communications]

A young child formula with *Limosilactobacillus reuteri* and GOS modulates gut microbiome and enhances bone and muscle development: A randomized trial.

Corresponding Author: Dr Marie-Noelle Horcajada

Version 0:

Reviewer comments:

Reviewer #1

(Remarks to the Author)

Summary and Noteworthy Results

This randomized, double-blind trial in 138 toddlers (2–3 y) compared an experimental young-child formula (EYCF; containing pre-cultivated *Limosilactobacillus reuteri* DSM 17938 + 4 g/L GOS) to a minimally fortified control milk (CM) over 6 months.

- Primary endpoint: Tibia speed of sound (SOS) increased by 58.9 m/s (1.5 %) in EYCF vs. CM ($p=0.002$) at 6 months, with a significant effect already at 3 months ($\Delta 53.0$ m/s; $p=0.024$).
- Secondary endpoint: Handgrip strength in the right arm was 11% higher in EYCF vs. CM at 6 months ($p=0.015$).
- Microbiome & metabolome: EYCF induced a bifidogenic effect (\uparrow *L. reuteri*, *Bifidobacterium* spp.) and altered fecal metabolites (e.g. \uparrow indole-3-propionic acid, lysine, pipecolic acid) that correlated with bone quality and muscle strength.
- Mechanistic in vitro data: Fermentate of EYCF enhanced myoblast fusion (+10%) and myotube area (+18%), and selected microbiota-derived metabolites (e.g. indole-3-carboxyaldehyde, hydroxyproline) stimulated osteoblast (ATF-4 expression) and muscle-progenitor parameters.

Significance and Novelty

- Importance to the field: Toddlerhood is a critical window for musculoskeletal and microbiome development; this is the first RCT demonstrating that a synbiotic formula may enhance ultrasound bone parameters in the tibia, but not in the radius, and muscle force in early life.
- Comparison to literature: Prior studies have shown either prebiotic or probiotic benefits on mineral absorption or bone in adults/rodents. Here, the combination shows synergistic effects in children, extending gut-bone and gut-muscle axis concepts to early development.
- Originality: The integration of clinical endpoints with shotgun metagenomics, untargeted metabolomics, and in-vitro functional assays offers a comprehensive mechanistic picture.

Support for Conclusions & Need for Additional Evidence

• Strengths:

- o Rigorous randomized, double-blind design with a non-fortified reference group.
- o Appropriate sample-size calculation for the primary outcome.
- o MMRM and FDR adjustments for repeated measures and multiple testing.
- o Concordant in vitro data supporting observed clinical associations.

• Limitations / Areas for further work:

- o Generalizability: Study population was exclusively Filipino; microbiome and growth responses may differ in other ethnicities.
- o Reference group design: The habitual-diet arm was non-randomized, introducing potential confounding despite propensity-score adjustment.
- o Robustness of the observed effects: The difference in SOS in the tibia was 1.5% in EYCF vs. CM, in the radius it was not significant. While the observed increases in SOS in EYCF were similar to normal age-associated increases in this age range, the lack of increases in CM and REF groups is rather unexpected. While baseline tibia SOS was included as a fixed covariate, a more rigorous analysis of the individual changes relative to baseline SOS (in both tibia and radius) would be desirable.
- o Dropout & compliance: Only 59% completed the intervention in each formula arm; implications for bias and power deserve

further discussion.

o Strain engraftment: Although *L. reuteri* increased, qPCR confirmation is needed to verify colonization by the DSM 17938 strain versus background.

o Clinical relevance: The magnitude of SOS increase (~2–3% per 6 months) should be contextualized in terms of fracture risk or long-term skeletal outcomes.

o Longer-term follow-up: It remains to be seen whether early gains in SOS and handgrip translate to persistent benefits into later childhood.

Data Analysis, Interpretation, and Conclusions

• Statistical methods are sound and appropriately reported; the use of MMRM, covariate adjustment, particularly adjustment for the expected age- and measurement associated increase of axial transmission SOS is commendable.

• SOS reference values of the Sunlight Omnisense exhibit a system-specific association with age (Zadik et al. 2003): The strong SOS gradient between 1 and 4 years is most likely associated with an increase of cortical thickness and degree of mineralization, while the SOS gradient between 4 – 12 years is associated with an increase in mineralization only, because i) the increase of cortical BMD is linear between 1 and 10 years (Rauch and Schonau, JBMR 16, Number 4, 2001), a linear increase in cortical thickness is expected between years 1 and 10 (Parfitt Osteoporosis Int, 1994), and ii) the velocity of the first arriving signal measured by the device is known to be affected by cortical thickness for thickness value smaller than the wavelength (Nieh et al. Medical Engineering & Physics 21, 1999), which can be explained by the transition of a guided wave mode (if the thickness is smaller than the wavelength) to a compressional wave (if the thickness is larger than the wavelength) (Raum et al. Ultrasound in Med. & Biol., Vol. 31, No. 9, 2005). The investigated cohort is within the age-range, in which changes in SOS are expected to be dominated by changes in cortical thickness:

Although both, increased cortical thickness and BMD (or subsequently matrix stiffness) result in higher bone quality during this stage of development, the observed changes do not allow a distinction, which bone quality property has improved. The plot above shows the data presented by the authors relative to reference data published by Zadik et al. (as the precise age of the Bonnet cohort was not known, a mean age of 2.5 years at baseline was assumed). Different points can be observed:

- The SOS values in the Bonnet study are generally lower than those of a Caucasian population published by Zadik et al., which may be due to ethnic / nutritional differences
- The radius values are generally lower than the tibia values those, which is not in line with the data published by Zadik et al.
- The strong age-dependence underlines the need of the analysis of relative increases compared to baseline measurements, rather than reporting the average values between baseline and follow-up time points
- All SOS values except for tibia REF and tibia CM follow the normal age-associated increase of SOS (which is approximately 100 m/s per year at the age between 2 and 3.5 years). Thus, it could be argued that nutrition in the REF and CM groups inhibit/delay normal growth in the tibia, but not in the radius, and the nutrition in the EYCF groups preserves normal bone development in the tibia only.
- Overall, the observed differences are very small compared to the precision of the SOS measurement and expected SOS increase within this observation window

- The correlation network linking taxa, metabolites, and clinical outcomes is well executed, but directionality cannot be inferred.
- Conclusions about gut-musculoskeletal crosstalk are justified, yet the manuscript should clarify that causality is inferred rather than proven.

Methodology and Reproducibility

- The Methods section provides detailed protocols for clinical measurements (SOS, handgrip), microbiome sequencing, metabolomics, and in vitro assays—including reagent sources, instrument models, and analysis pipelines. However, the SOS measurements should be described in more detail (repeated measurements? Reproducibility)
- Randomization, blinding, and sample-size rationale are clearly described.
- All assays appear reproducible, though deposition of raw sequence and metabolomics data in public repositories would strengthen transparency.

Detailed Comments

P4I109: The sentence should be rephrased. Replace “drastically” by a more descriptive wording and “is highly dependent on nutrition” should be accompanied by other factors influencing muscle strength and mass

P5I122: the primary endpoint “tibia quality (ultrasound)” should be defined more precisely

P6I148: what is “FAS analysis?”

P13I328: “... exhibited a gain in tibia SOS by 4 times ...” would assume that SOS is invariant within the 6-months follow-up time, which is not the case, e.g. all radius measurements exhibit the same age-associated increase

P13I332: “... improving bone and muscle strength acquisition”. Bone strength was not measured

P13I335: the increase in SOS need to be discussed and interpreted, as explained above.

Supplementary Table formatting

Fig.1 A, capture and p6I148 and : What is FAS analysis?

Fig.1 C,D, and capture: The capture reports that “Data are presented as mean ± SD”, but SDs of SOS are in the > 100 m/s range, while the error bars are much smaller. This needs to be corrected for this (and potentially for all figures)

Fig. 1 C,D: outliers in C (above 3600 m/s and below 3300 m/s) cannot be seen in D, as the axis limits are much narrower

Suppl. Table S2: very difficult to read, as table columns are split over several pages, units for tibia and radius SOS, length, handgrip and bone turnover are missing

Overall Recommendation:

This manuscript reports novel and clinically relevant findings with sound methodology. I recommend major revision to address the points above—particularly discussing generalizability, dropout impact, strain-specific colonization, and clinical

significance of the observed effects.

Reviewer #2

(Remarks to the Author)

The manuscript by Bonnet et al. entitled "A young child formula with *L. reuteri* and GOS enhances bone and muscle development via microbiome" describes the outcomes of a clinical study in young children (2-3 years) that received for 6 months minimally fortified milk (control group) or a formula that contains a mixture of *L. reuteri* DSM 17938 fortified with GOS (experimental group). The main outcomes reported were changes in the microbiome (increased Bifidobacteria and *L. reuteri*), higher bone quality and length and changes in the metabolome. The final outcomes in bone health / muscle strength are impressive. Expansion of the microbiome, especially, *L. reuteri*, was as expected as this was supplemented. Major changes observed in metabolites may be attributed to overall microbiome changes but not directly linked to the observed bone phenotypes. Causal evidence is lacking. Also, it is unclear if *L. reuteri* or GOS alone would impact bone health.

General Comments:

The title should be changed. It implies that the microbiome drives enhancement in bone and muscle development following the administration of *L. reuteri* and GOS. There is no causal evidence presented to support that. Changes are observed but these may have nothing to do with the observed phenotypes. The only conclusion that can be drawn is that *L. reuteri* DSM17938 +GOS changes the microbiome, and no mechanisms can be only be speculated at this point.

The introduction is choppy, and could be improved for flow. The introduction of GOS (which should be explained to the broad audience what this is) flows into experimental background that does not involve GOS, and then it jumps back to referring to GOS, for example. It helps if the introduction flows from general background to the gap in the knowledgebase to the pertaining study.

L107: "...suggesting that probiotics are a novel candidate approach to prevent age-associated bone loss and osteoporosis." Change to: "...suggesting that the application select probiotic strains could be a novel approach to prevent age-related bone health and osteoporosis". The critical part here is to refer to "select strains" and not to place this phenotype under the umbrella of 'probiotics' (or microbiome in general). I encourage the authors to apply this throughout the manuscript.

L117-118: "remains unknown whether modulating the microbiome can impact...". This should be changed to: "remains unknown whether administration of select probiotic strains can impact..."

The preconditioning of *L. reuteri* with GOS seems to be important, and this should be explained in detail of what this entails. Metabolomic studies from cultures of *L. reuteri* that have and have not been preconditioned with GOS provided more insight into which

L361: there should be discussion regards the observations of changes in metabolites related to Trp metabolism, how this can be linked to *L. reuteri*. There are numerous studies now that show a role of *L. reuteri* to convert Trp to I3C, for example, and the role it has in immunomodulation. See, for example, pioneering work by Zelante et al. (2013) and multiple studies that followed by others.

Reviewer #3

(Remarks to the Author)

This is a randomized, double-blind controlled trial of toddlers investigating the effects of adding *Lactobacillus Reuteri* and GOS to a toddler's milk (EYCF) during a 6-month period in comparison to a feeding a minimally fortified milk (CM), with a primary aim to investigate the effects on bone strength. In addition, this study also looked on the effects on muscle strength. Furthermore, effects on the microbiome and the fecal metabolome were explored and associations to bone quality and muscle strength investigated. The authors found an increase in tibia speed of sound and a higher muscle strength in the EYCF group, changes in the microbiome composition as well as in the fecal metabolome which were associated to the intervention.

The study adds new and important information to this scientific field. The study is well performed, the aims clear, the methods thoroughly described, and the analyses appropriately done. Different aspects of the results are suitably presented and appropriately discussed in the discussion part. The paper is also overall well written.

I have a few aspects to be commented on:

1. When calculating the sample size needed to achieve an overall type I error of 5% and 90% statistical power, 77 children completing the study were needed. However, only 59% completed the 6-month period. How have the authors approached this fact in their statistical calculations? Should this high drop-out rate have influenced the interpretation of the results?
2. Not all children in the EYCF group had increases *Lact Reuteri* in the microbiome of, despite intake of the fortified milk. Why is that?
3. Why did not both experimental formulas have the same vitamin D concentrations? Could this be a problem when comparing the two groups?
4. I would suggest that the units of Vitamin D are presented in the same way in both experimental formulas

Version 1:

Reviewer comments:

Reviewer #1

(Remarks to the Author)

The authors have adequately addressed the majority of my comments.
There are a few points that still need to be addressed:

1. Figure Quality: Fonts of Figs. 1, 2 are too small to be readable. They should be revised
2. Page 4 Line: 123: "tibia speed of sound (index of bone quality integrating bone density, trabecular & cortical architecture and geometry)" is too vague. tibia speed of sound (or more precisely the velocity of the first arriving signal) measured by axial transmission ultrasound, is associated with sound velocity in the cortical shell (which is determined by matrix mineralisation, tissue stiffness and porosity) and cortical thickness (only for small thickness values), but not trabecular architecture and geometry (see also my initial comments). The sentence should be revised accordingly.
3. Page 13 Line 330: "improved" should be "increased" and "tibia SOS higher by 4 times" needs to be rephrased, as SOS was not 4 times higher
3. Page 14 Line 335: "improving bone quality" needs to be rephrased as SOS is not bone quality. I suggest a clear definition of which aspects of bone quality tibia SOS is associated with and this definition is then referred to throughout the manuscript
4. Page 25 Section speed of sound of the tibia and radius: To shorten the manuscript, while still maintaining essential details, I suggest moving the majority of detailed description (also what has been provided in the rebuttal) to supplementary materials, while including the information about reproducibility to the manuscript text body. Given the small observed changes in relation to the respective CVs, this information must not be omitted.

Typos:

P395: should read "However, ..."

P14|339: Sos should be SOS

P39 |965 and |969: "is be included in order to" needs to be rephrased

Reviewer #2

(Remarks to the Author)

The authors have addressed all my comments/concerns.

Reviewer #3

(Remarks to the Author)

The authors have responded satisfactorily to my comments.

REVIEWER EXPERTISE

Reviewer #1. Quantitative ultrasound methods in bone, clinical studies

Reviewer #2. Probiotics, microbiome

Reviewer #3. Infant Formula, infant nutrition, clinical trials

Points by point answers have been made in blue.

REVIEWER COMMENTS

Reviewer #1 (Remarks to the Author):

Summary and Noteworthy Results

This randomized, double-blind trial in 138 toddlers (2–3 y) compared an experimental young-child formula (EYCF; containing pre-cultivated *Limosilactobacillus reuteri* DSM 17938 + 4 g/L GOS) to a minimally fortified control milk (CM) over 6 months.

- Primary endpoint: Tibia speed of sound (SOS) increased by 58.9 m/s (1.5 %) in EYCF vs. CM ($p=0.002$) at 6 months, with a significant effect already at 3 months ($\Delta 53.0$ m/s; $p=0.024$).
- Secondary endpoint: Handgrip strength in the right arm was 11% higher in EYCF vs. CM at 6 months ($p=0.015$).
- Microbiome & metabolome: EYCF induced a bifidogenic effect (\uparrow *L. reuteri*, *Bifidobacterium* spp.) and altered fecal metabolites (e.g. \uparrow indole-3-propionic acid, lysine, pipecolinic acid) that correlated with bone quality and muscle strength.
- Mechanistic in vitro data: Fermentate of EYCF enhanced myoblast fusion (+10%) and myotube area (+18%), and selected microbiota-derived metabolites (e.g. indole-3-carboxyaldehyde, hydroxyproline) stimulated osteoblast (ATF-4 expression) and muscle-progenitor parameters.

Significance and Novelty

- Importance to the field: Toddlerhood is a critical window for musculoskeletal and microbiome development; this is the first RCT demonstrating that a synbiotic formula may enhance ultrasound bone parameters in the tibia, but not in the radius, and muscle force in early life.
- Comparison to literature: Prior studies have shown either prebiotic or probiotic benefits on mineral absorption or bone in adults/rodents. Here, the combination shows synergistic effects in

children, extending gut-bone and gut-muscle axis concepts to early development.

- Originality: The integration of clinical endpoints with shotgun metagenomics, untargeted metabolomics, and in-vitro functional assays offers a comprehensive mechanistic picture.

Support for Conclusions & Need for Additional Evidence

- Strengths:

- o Rigorous randomized, double-blind design with a non-fortified reference group.

- o Appropriate sample-size calculation for the primary outcome.

- o MMRM and FDR adjustments for repeated measures and multiple testing.

- o Concordant in vitro data supporting observed clinical associations.

Thanks for your valuable comments.

- Limitations / Areas for further work:

- o Generalizability: Study population was exclusively Filipino; microbiome and growth responses may differ in other ethnicities.

We agree with the reviewer that the generalizability of our findings beyond the Filipino population is an important consideration. Conducting the study in a single geography was intentional, to minimize potential confounding effects due to ethnic and dietary heterogeneity that could have an impact in the microbiome of weaning toddlers. To acknowledge this, we have explicitly added this point in our manuscript l. 340-344. We also would like to note that the control arm provides an internal benchmark, and the observed differences were consistent within this population. In addition, *in vitro* experiments conducted as part of this research provide experimental evidence under controlled, population-independent conditions.

- o Reference group design: The habitual-diet arm was non-randomized, introducing potential confounding despite propensity-score adjustment

Thank you for your comments. The section “statistical analysis on primary and secondary outcomes” describes our strategy for handling the non-randomized nature of the third arm. Specifically, lines 744-753 outline the use of propensity score.

- o Robustness of the observed effects: The difference in SOS in the tibia was 1.5% in EYCF vs. CM, in the radius it was not significant. While the observed increases in SOS in EYCF were

similar to normal age-associated increases in this age range, the lack of increases in CM and REF groups is rather unexpected. While baseline tibia SOS was included as a fixed covariate, a more rigorous analysis of the individual changes relative to baseline SOS (in both tibia and radius) would be desirable.

Thank you for your comments. We agree that during the 6 months period of our experiment, Filipino control and reference groups do not increase their tibia Sos as expected based on literature (1). As described in the materials and methods, 70% of Filipino toddler exhibit vitamin D insufficiency, known to be key for tibia bone quality set-up (SOS) (2). This may explain the slow rate of tibia Sos gain in CM and REF groups. We would like to emphasize that the three groups have a similar rate of tibia length increase, mirroring a normal growth. As you notice, EYCF group does not significantly differ from the two other groups at the radius site, a non-loading bone, suggesting a potential interaction of our nutritional intervention with mechanical loading, e.g weight bearing bone. Such observation have also been made with calcium supplementation showing an efficacy mainly on the loaded bone and not the unloaded bone (3). Further no difference was observed in daily recreational activities measured using the 3-day Outdoor Playtime Checklist administered prior to the visits among the 3 groups (data not shown in the publication but can be consulted in the statistical report section 7.3.2 “physical parameters” joined to the submission).

Concerning suggestion to perform an analysis of the individual changes relative to baseline SOS (in both tibia and radius), from a statistical point of view, this analysis would be equivalent to the post baseline levels corrected for baseline level (results are the same, only the scale changes).

On top of figure 1D please see below for information individual changes of tibia and radius SoS.

Trajectories of the tibia SOS Measurements (ITT)

Trajectories of the radius SOS Measurements (ITT)

o Dropout & compliance: Only 59% completed the intervention in each formula arm; implications for bias and power deserve further discussion.

Thank you for this comment. The statement “A total of 107 (59%) subjects completed the 6 months intervention period in the study, of whom 51 had been randomized to EYCF and 56 to CM” is incomplete and should read as follows: “A total of 107 (59%) subjects completed the 6 months

intervention period in the per protocol population, of whom 51 had been randomized to EYCF and 56 to CM". As described in Figure 1, 225 (82%) subjects completed the study and the retention rate in the FAS was 77% in both experimental arms. This has been corrected in the manuscript (1154-157). The retention rate observed in this study aligns with those reported in previously published studies involving milk administration of comparable duration and conducted in populations of similar age (4, 5, 6, 7) .

Further, we do believe that our comparative conclusions between randomized arms are unaffected by this higher than anticipated loss of information because the induced missing data appears to be missing at random. Especially it is likely to be independent from the allocation as demonstrated by the even number of subjects missing in each arms and the reasons reported (taste of the products, transfer to other province / country and relatives disapproval are the main reason for dropping out of the trial).

o **Strain engraftment:** Although *L. reuteri* increased, qPCR confirmation is needed to verify colonization by the DSM 17938 strain versus background.

We agree that our analysis does not allow to definitively distinguish whether the observed increase in *L. reuteri* originates from the administered intervention strain or from background colonization. For this reason, we were cautious to report changes at the species level only, not expanding the interpretation to strain level, and avoided making claims about probiotic engraftment. As noted in the manuscript (lines 416-422), we also acknowledged this as a limitation of the study: "*Further analysis using qPCR is needed to confirm the origin of the L. reuteri strain, i.e., from the synbiotic intervention or from other sources, as some samples in the control group (3 out of 69 participants) also showed an increase in L. reuteri abundance.*"

o **Clinical relevance:** The magnitude of SOS increase (~2–3% per 6 months) should be contextualized in terms of fracture risk or long-term skeletal outcomes.

Unfortunately contextualization of the magnitude of SOS changes and impact on fracture risk in toddler population is not possible due to lack of existing data. However, in adults we know that 100 m/s decrease in SOS increases fracture odds ratio by 50% (8). Osteoporotic treatment like Alendronate known to decrease fracture risk by 55% (9) and was also demonstrated to induce a significant change on tibia SoS after 6 months of treatment with mean difference of 57.4m/s

compared to placebo (10). In our study, tibia speed of sound increased by 58.9 m/s in EYCF vs. CM ($p=0.002$) at 6 months, suggesting that our Experimental Young formula are closed to Alendronate efficacy. Combining all these information, we suggest that our formula would impact fracture risk and could be an interesting strategy for osteoporotic subjects. However this statement is quite speculative, that is why we decided not to include it in the discussion of the manuscript. Same applies for long term impact: it is very challenging to look into adult space (where data are scarce) and speculate for toddlers. Ideally a follow-up measure of the subjects after peak bone mass achievement would be required to answer this point to conclude on long term skeletal impact of our intervention.

o Longer-term follow-up: It remains to be seen whether early gains in SOS and handgrip translate to persistent benefits into later childhood.

We fully agree with the very interesting opportunity it would bring to have a follow up analysis to check if early gains in SOS and handgrips translate to persistent benefits into later childhood, however, unfortunately this has not been planned in the protocol of our study. We added following statement “Whether early gains in bone quality and muscle strength translate to persistent benefits into later childhood remains to be demonstrated. “ in the conclusion of the manuscript (l. 437-438).

Data Analysis, Interpretation, and Conclusions

- Statistical methods are sound and appropriately reported; the use of MMRM, covariate adjustment, particularly adjustment for the expected age- and measurement associated increase of axial transmission SOS is commendable.
- SOS reference values of the Sunlight Omnisense exhibit a system-specific association with age (Zadik et al. 2003):

The strong SOS gradient between 1 and 4 years is most likely associated with an increase of cortical thickness and degree of mineralization, while the SOS gradient between 4 – 12 years is associated with an increase in mineralization only, because i): the increase of cortical BMD is linear between 1 and 10 years (Rauch and Schonau, JBMR 16, Number 4, 2001), a linear increase in cortical thickness is expected between years 1 and 10 (Parfitt Osteoporosis Int, 1994), and ii) the velocity of the first arriving signal measured by the device is known to be affected by cortical thickness for thickness value smaller than the wavelength (Nieh et al. Medical

Engineering & Physics 21, 1999), which can be explained by the transition of a guided wave mode (if the thickness is smaller than the wavelength) to a compressional wave (if the thickness is larger than the wavelength) (Raum et al. Ultrasound in Med. & Biol., Vol. 31, No. 9, 2005).

The investigated cohort is within the age-range, in which changes in SOS are expected to be dominated by changes in cortical thickness:

Although both, increased cortical thickness and BMD (or subsequently matrix stiffness) result in higher bone quality during this stage of development, the observed changes do not allow a distinction, which bone quality property has improved. The plot above shows the data presented by the authors relative to reference data published by Zadik et al.. (as the precise age of the Bonnet cohort was not known, a mean age of 2.5 years at baseline was assumed). Different points can be observed:

- The SOS values in the Bonnet study are generally lower than those of a Caucasian population published by Zadik et al., which may be due to ethnic / nutritional differences
- The radius values are generally lower than the tibia values those, which is not in line with the data published by Zadik et al.

We fully agree with these 2 points above, and, as you rightly mentioned, could be linked to ethnicity, nutritional and/or level of physical activities differences between the 2 studies. However we believe these points are not to be discussed in the manuscript, to keep the flow of our paper and would not bring new elements regarding the conclusions on efficacy of our EYCF.

- The strong age-dependence underlines the need of the analysis of relative increases compared to baseline measurements, rather than reporting the average values between baseline and follow-up time points

"...we show that if one uses an analysis of covariance on the gain scores, using the pretest as the covariate, the treatment comparisons will be identical to the treatment comparisons from an analysis of covariance on the posttest scores." (11).

- All SOS values except for tibia REF and tibia CM follow the normal age-associated increase of SOS (which is approximately 100 m/s per year at the age between 2 and 3.5 years). Thus, it could be argued that nutrition in the REF and CM groups inhibit/delay normal growth in the tibia, but not in the radius, and the nutrition in the EYCF groups preserves normal bone

development in the tibia only.

- Overall, the observed differences are very small compared to the precision of the SOS measurement and expected SOS increase within this observation window.

We agree that the difference reported can be seen as small. However such results need to be conceptualized specifically in bone field where small effects on bone quantity and/or quality can translate into significant effect in term of fracture risk See above example mentioned with alendronate drug (10) (12). On top if we compare to existing nutritional solutions we can see that the effect size is in the same range:

- +2.5% higher tibia SoS after 3 months of high vs low beta-palmitate levels in infant formulae, $P=0.04$ (13).
- +0.8% higher metacarpal Sos after 12 months of vitamin D supplementation versus no supplementation, $p<0.001$ (2).

Further, at this age, it has been reported that toddlers have an increase of tibia Sos around 2.1% per year (1). This suggest that the that the increase of 1.5% observed in our study after 6 months would translate into a gain in bone maturation time of 8.5 months.

This has been added in the discussion section l. 338-344.

- The correlation network linking taxa, metabolites, and clinical outcomes is well executed, but directionality cannot be inferred.

We fully agree with the reviewer that directionality in the sense of causation cannot be inferred from correlations and we were careful throughout the manuscript not to imply causality or directional effects (see changes paragraph “The experimental young child formula impacts microbiome species & metabolites that are associated to clinical outcomes” l. 238-286). In response to this comment, we re-reviewed the relevant sections and made minor edits where needed to ensure that all statements about correlations remain appropriately cautious and neutral regarding directionality (l. 238-239; 248-249; l. 266). Of note, this is also why our clinical results are complemented by *in vitro* experiments of selected metabolites on osteoblast and muscle progenitor cell models, assessing key markers of bone and muscle development (l. 276-286). We believe that these experiments support the plausibility of the observed associations between metabolites and clinical outcomes, without making any statements on directionality.

- Conclusions about gut-musculoskeletal crosstalk are justified, yet the manuscript should clarify that causality is inferred rather than proven.

We fully agree with the reviewer and amend the discussion as mentioned in above answer.

Methodology and Reproducibility

- The Methods section provides detailed protocols for clinical measurements (SOS, handgrip), microbiome sequencing, metabolomics, and in vitro assays—including reagent sources, instrument models, and analysis pipelines. However, the SOS measurements should be described in more detail (repeated measurements? Reproducibility)

Speed of ultrasound was measured at the distal third of the radius and proximal third of the tibia, using the Sunlight Omnisense® 9000S Bone Sonometer (BeamMed Ltd) according to the manufacturer recommendations. The Sunlight Omnisense® is a portable ultrasonometer which emits and receives pulses of ultrasound along the radius bone at low 200-kHz frequency and measures the speed of sound in the radius (low-frequency velocity, VLF, m/s). The probe consists of an array of source and receiver elements operating according to a bidirectional principle so as to accurately correct for the effects of overlying soft tissue. Ultrasound gel was used as the contact medium. Once the probe was positioned, firm pressure (approx. 1 kg) was applied against the skin. The gel and the pressure applied ensured that optimum acoustic coupling was achieved so that soft tissue effects were minimized (14). Two trained investigators worked on VLF assessments and performed two measurements on four same participants.

	Investigator 1	Investigator 2	Interobserver
Coefficient of variation radius	4.43%	3.80%	4.16%
Coefficient of variation tibia	1.16%	1.82%	2.07%

We did not include all above detailed methodology (and coefficient of variation achieved in this study) regarding bone SoS measurements because of manuscript length that is already to be shorten. Further, publications with same device used are available and cited in the method section line 613-616 (1).

- Randomization, blinding, and sample-size rationale are clearly described.
- All assays appear reproducible, though deposition of raw sequence and metabolomics data in public repositories would strengthen transparency.

Metabolomics data has been submitted to MetaboLights (accession MTBLS12557) and is available to reviewers upon request. Similarly, the metagenomic data has been uploaded to ENA (accession RJEB83333). Both datasets will be made publicly available upon publication of the manuscript.

Detailed Comments

P41109: The sentence should be rephrased. Replace “drastically” by a more descriptive wording and “is highly dependent on nutrition” should be accompanied by other factors influencing muscle strength and mass

This has been addressed, see l. 111-112.

P51122: the primary endpoint “tibia quality (ultrasound)” should be defined more precisely

We have precise that the primary endpoint is the tibia SOS which is an index of bone quality integrating density, trabecular & cortical architecture and geometry.

P61148: what is “FAS analysis?”

FAS analysis stands for Full analysis set and has been added line 151.

P131328: “... exhibited a gain in tibia SOS by 4 times ...” would assume that SOS is invariant within the 6-months follow-up time, which is not the case, e.g. all radius measurements exhibit the same age-associated increase

Rewording as follows has been made: “... exhibited a tibia SOS higher by 4 times compared to the habitual diet group” line 333.

P131332: “... improving bone and muscle strength acquisition”. Bone strength was not measured

Reworded as follow: “... improving bone quality and muscle strength acquisition” see line 336.

P131335: the increase in SOS need to be discussed and interpreted, as explained above.

Please see explanation to your previous comment on small effect size. Discussion has been amended accordingly.

Supplementary Table formatting

Fig.1 A, capture and p61148 and : What is FAS analysis?

FAS refer to Full analysis set. Capture in Fig. 1A legend

Fig.1 C,D, and capture: The capture reports that “Data are presented as mean \pm SD“, but SDs of SOS are in the > 100 m/s range, while the error bars are much smaller. This needs to be corrected for this (and potentially for all figures)

We apologize for the mistake, only Fig.1 C, E are displaying SD. Correction has been done in the figure legend of Figure 1 (line 824).

Fig. 1 C,D: outliers in C (above 3600 m/s and below 3300 m/s) cannot be seen in D, as the axis limits are much narrower

We have adjust the axis of figure D as in figure C, however we lose visualization of differences between YCF and CM (see below). If you feel that we should keep same Y axis scale will do, however outliers will remain not visible since it is not an individual representation plot.

Suppl. Table S2: very difficult to read, as table columns are split over several pages, units for tibia and radius SOS, length, handgrip and bone turnover are missing

We apologize for the difficulty to read the Suppl. Table S2. We have generated now a 1-page table with all units which we expect to be easier to read

Overall Recommendation:

This manuscript reports novel and clinically relevant findings with sound methodology. I recommend major revision to address the points above—particularly discussing generalizability, dropout impact, strain-specific colonization, and clinical significance of the observed effects.

This has been done throughout the discussion, aligned with the modifications mentioned above.

Reviewer #2 (Remarks to the Author):

The manuscript by Bonnet et al. entitled “A young child formula with *L. reuteri* and GOS enhances bone and muscle development via microbiome” describes the outcomes of a clinical

study in young children (2-3 years) that received for 6 months minimally fortified milk (control group) or a formula that contains a mixture of *L. reuteri* DSM 17938 fortified with GOS (experimental group). The main outcomes reported were changes in the microbiome (increased Bifidobacteria and *L. reuteri*), higher bone quality and length and changes in the metabolome. The final outcomes in bone health / muscle strength are impressive. Expansion of the microbiome, especially, *L. reuteri*, was as expected as this was supplemented. Major changes observed in metabolites may be attributed to overall microbiome changes but not directly linked to the observed bone phenotypes. Causal evidence is lacking. Also, it is unclear if *L. reuteri* or GOS alone would impact bone health.

General Comments:

The title should be changed. It implies that the microbiome drives enhancement in bone and muscle development following the administration of *L. reuteri* and GOS. There is no causal evidence presented to support that. Changes are observed but these may have nothing to do with the observed phenotypes. The only conclusion that can be drawn is that *L. reuteri* DSM17938 +GOS changes the microbiome, and no mechanisms can be only be speculated at this point.

We agree and have proposed the following title: “A young child formula with *L. reuteri* and GOS modulates gut microbiome and enhances bone and muscle development”

The introduction is choppy, and could be improved for flow. The introduction of GOS (which should be explained to the broad audience what this is) flows into experimental background that does not involve GOS, and then it jumps back to referring to GOS, for example. It helps if the introduction flows from general background to the gap in the knowledgebase to the pertaining study.

We have reshaped the introduction section to improve flow.

L107: “...suggesting that probiotics are a novel candidate approach to prevent age-associated bone loss and osteoporosis.” Change to: “...suggesting that the application select probiotic strains could be a novel approach to prevent age-related bone health and osteoporosis”. The critical part here is to refer to “select strains” and not to place this phenotype under the umbrella

of ‘probiotics’ (or microbiome in general). I encourage the authors to apply this throughout the manuscript.

Thanks for the recommendation that we have implemented. See l. 98-102.

L117-118: “remains unknown whether modulating the microbiome can impact...”. This should be changed to: “remains unknown whether administration of select probiotic strains can impact...” Thanks for the recommendation that we have implemented. See l.100-102.

The preconditioning of *L. reuteri* with GOS seems to be important, and this should be explained in detail of what this entails. Metabolomic studies from cultures of *L. reuteri* that have and have not been preconditioned with GOS provided more insight into which

Preconditioning of *L. reuteri* has been already described in our previous paper published in Scientific reports (15). In summary the concept is to prime the bacterial enzymatic machinery of a probiotic strain for a specific prebiotic here GOS by having a structurally similar substrate as a carbon source during production. We demonstrated that preconditioning leads to improved ex vivo metabolic profiles, microbial community dynamics, probiotic engraftment, and ex vivo osteoblastogenesis (15). This point has been added l. 585-589.

L361: there should be discussion regards the observations of changes in metabolites related to Trp metabolism, how this can be linked to *L. reuteri*. There are numerous studies now that show a role of *L. reuteri* to convert Trp to I3C, for example, and the role it has in immunomodulation. See, for example, pioneering work by Zelante et al. (2013) and multiple studies that followed by others.

We thank the reviewer for pointing this out, as there is indeed evidence showcasing the link between *L. reuteri* and Trp metabolism, particularly concerning indole-3-carboxyaldehyde (indole-3-aldehyde, I3C). We have added this point in the discussion as well as the reference (l. 370-374).

Reviewer #3 (Remarks to the Author):

This is a randomized, double-blind controlled trial of toddlers investigating the effects of adding

Lactobacillus Reuteri and GOS to a toddler's milk (EYCF) during a 6-month period in comparison to a feeding a minimally fortified milk (CM), with a primary aim to investigate the effects on bone strength. In addition, this study also looked on the effects on muscle strength. Furthermore, effects on the microbiome and the fecal metabolome were explored and associations to bone quality and muscle strength investigated. The authors found an increase in tibia speed of sound and a higher muscle strength in the EYCF group, changes in the microbiome composition as well as in the fecal metabolome which were associated to the intervention. The study adds new and important information to this scientific field. The study is well performed, the aims clear, the methods thoroughly described, and the analyses appropriately done. Different aspects of the results are suitably presented and appropriately discussed in the discussion part. The paper is also overall well written.

I have a few aspects to be commented on:

1. When calculating the sample size needed to achieve an overall type I error of 5% and 90% statistical power, 77 children completing the study were needed. However, only 59% completed the 6-month period. How have the authors approached this fact in their statistical calculations? Should this high drop-out rate have influenced the interpretation of the results?

Thank you for your comments. The number 59% corresponds to the number of subjects who completed the 6 months visit without having any major protocol deviation leading to exclusion from the per protocol set. The number of subjects with primary endpoint data available at 6 months in the full analysis set (FAS) was 69 in each randomized arms, which is close to our initial target. We do believe that our comparative conclusions between randomized arms are unaffected by this higher than anticipated loss of information because the induced missing data appears to be missing at random. Especially it is likely to be independent from the allocation as demonstrated by the even number of subjects missing in each arms and the reasons reported (taste of the products, transfer to other province / country and relatives disapproval are the main reason for dropping out of the trial).

2. Not all children in the EYCF group had increases Lact Reuteri in the microbiome of, despite intake of the fortified milk. Why is that?

We thank the reviewer for this observation and agree that not all children in the EYCF group showed an increase in *L. reuteri* at the time points assessed. This type of inter-individual variation

is inherent to human intervention studies and is commonly observed even in relatively homogeneous populations. We would like to note that the absence of detectable *L. reuteri* in some participants does not necessarily imply absence of response to the intervention, as functional effects may occur through transient passage, modulation of microbial networks, or downstream metabolic changes rather than persistent colonization. Previous studies have indeed shown that engraftment is not per se required to elicit functional or clinical effects (see Suez et al. 2019 <https://www.nature.com/articles/s41591-019-0439-x>) We have added a sentence to the discussion acknowledging this point (l. 416-419).

3. Why did not both experimental formulas have the same vitamin D concentrations? Could this be a problem when comparing the two groups?

Thank you for this comment. The primary objective of the study was to assess the effect of an experimental young child formula (EYCF) containing a synbiotic on bone quality, in comparison to a minimally fortified milk (CM). The EYCF, including its nutritional composition, was formulated in accordance with the latest regulatory standards (fao.org/fao-who-codexalimentarius/sh-proxy/es/?lnk=1&url=https%253A%252F%252Fworkspace.fao.org%252Fsites%252Fcodex%252FStandards%252FCXS%2B156-1987%252FCXS_156e.pdf). In contrast, the CM was designed to resemble powdered cow's milks commonly consumed by young children in the Philippines, which typically have lower vitamin D concentrations.

The impact of the nutritional interventions on vitamin D (25-hydroxycholecalciferol) levels was evaluated. After 6 months of intervention, vitamin D levels were significantly higher in the EYCF compared to CM. However, blood vitamin D levels were not associated to tibia SOS and muscle force as presented in Figures 2B-2C.

4. I would suggest that the units of Vitamin D are presented in the same way in both experimental formulas

Thank you for this very valid comment. We corrected it in the Supplemental Table S16 and the vitamin D levels of both products are now expressed in $\mu\text{g}/100\text{g}$.

References

1. Zadik Z, Price D, Diamond G. Pediatric reference curves for multi-site quantitative ultrasound and its modulators. *Osteoporos Int* 2003;14(10):857-62.
2. Savino F, Viola S, Tarasco V, Lupica MM, Castagno E, Oggero R, Miniero R. Bone mineral status in breast-fed infants: influence of vitamin D supplementation. *Eur J Clin Nutr* 2011;65(3):335-9.
3. Courteix D, Jaffre C, Lespessailles E, Benhamou L. Cumulative effects of calcium supplementation and physical activity on bone accretion in premenarchal children: a double-blind randomised placebo-controlled trial. *Int J Sports Med*. 2005;26((5)):332-8.
4. Rivera-Pasquel M, Flores-Aldana M, Parra-Cabrera MS, Quezada-Sánchez AD, García-Guerra A, Maldonado-Hernández J. Effect of Milk-Based Infant Formula Fortified with PUFAs on Lipid Profile, Growth and Micronutrient Status of Young Children: A Randomized Double-Blind Clinical Trial. *Nutrients*. 2020;13(1):doi: 10.3390.
5. Lovell AL, Davies PSW, Hill RJ, Milne T, Matsuyama M, Jiang Y, et al. Compared with Cow Milk, a Growing-Up Milk Increases Vitamin D and Iron Status in Healthy Children at 2 Years of Age: The Growing-Up Milk-Lite (GUMLi) Randomized Controlled Trial. *J Nutr* 2018 148(10):1570-9.
6. Akkermans MD, Eussen SR, van der Horst-Graat JM, van Elburg RM, van Goudoever JB, Brus F. A micronutrient-fortified young-child formula improves the iron and vitamin D status of healthy young European children: a randomized, double-blind controlled trial. *Am J Clin Nutr*. 2017;105(2):391-9.
7. Nocerino R, Paparo L, Terrin G, Pezzella V, Amoroso A, Cosenza L, et al. Cow's milk and rice fermented with *Lactobacillus paracasei* CBA L74 prevent infectious diseases in children: A randomized controlled trial. *Clin Nutr*. 2017;36(1):118-25.
8. Weiss M, Ben-Shlomo A, Hagag P, Ish-Shalom S. Discrimination of proximal hip fracture by quantitative ultrasound measurement at the radius. *Osteoporos Int* 2000;11(5):411-6.
9. Iwamoto J, Sato Y, Takeda T, Matsumoto H. Hip fracture protection by alendronate treatment in postmenopausal women with osteoporosis: a review of the literature. *Clinical interventions in aging*. 2008;3(3):483-9.

10. Drake WM, Brown JP, Banville C, Kendler DL. Use of phalangeal bone mineral density and multi-site speed of sound conduction to monitor therapy with alendronate in postmenopausal women. *Osteoporos Int* 2002;13(3):249-56.
11. Laird N. Further Comparative Analyses of Pretest-Posttest Research Designs. *The American Statistician*. 1983;37(4a):329–30.
12. Olszynski WP, Davison KS, Adachi JD, Brown JP, Hanley DA. Change in Quantitative Ultrasound-assessed Speed of Sound as a Function of Age in Women and Men and Association With the Use of Antiresorptive Agents: The Canadian Multicentre Osteoporosis Study. *J Clin Densitom*. 2020;23(4):549-60.
13. Litmanovitz I, Davidson K, Eliakim A, Regev RH, Dolfin T, Arnon S, et al. High Beta-palmitate formula and bone strength in term infants: a randomized, double-blind, controlled trial. *Calcif Tissue Int*. 2013;92(1):35-41.
14. Biver E, Pepe J, de Sire A, Chevalley T, Ferrari S. Associations between radius low-frequency axial ultrasound velocity and bone fragility in elderly men and women. *Osteoporos Int*. 2019;30(2):411-21.
15. De Bruyn F, Bonnet N, Baruchet M, Sabatier M, Breton I, Bourqui B, et al. Galacto-oligosaccharide preconditioning improves metabolic activity and engraftment of *Limosilactobacillus reuteri* and stimulates osteoblastogenesis ex vivo. *Sci Rep*. 2024;14(4329):doi: 10.1038/s41598-024-54887-z.

REVIEWERS' COMMENTS

Reviewer #1 (Remarks to the Author):

The authors have adequately addressed the majority of my comments.
There are a few points that still need to be addressed:

1. Figure Quality: Fonts of Figs. 1, 2 are too small to be readable. They should be revised

Font size has been increased.

2. Page 4 Line: 123: "tibia speed of sound (index of bone quality integrating bone density, trabecular & cortical architecture and geometry" is too vague. tibia speed of sound (or more precisely the velocity of the first arriving signal) measured by axial transmission ultrasound, is associated with sound velocity in the cortical shell (which is determined by matrix mineralisation, tissue stiffness and porosity) and cortical thickness (only for small thickness values), but not trabecular architecture and geometry (see also my initial comments). The sentence should be revised accordingly.

Thanks for this valuable comment. The sentence has been revised accordingly with also added following reference: Schneider J, Iori G, Ramiandrisoa D, Hammami M, Gräsel M, Chappard C, et al. Ex vivo cortical porosity and thickness predictions at the tibia using full-spectrum ultrasonic guided-wave analysis. Arch Osteoporos 2019;14(1):21.
. See p4 line 124.

3. Page 13 Line 330: "improved" should be "increased" and "tibia SOS higher by 4 times" needs to be rephrased, as SOS was not 4 times higher

We agree with this comment; our sentence was not clear enough. We rephrased as follow: "We found that this formula increased tibia SOS compared to a minimally fortified control milk (CM) at 6 months timepoint (+1.5% vs CM). In addition, tibia SOS gain over 6 months was 4 times higher in the EYCF group compared to habitual diet (+1,7% vs +0.4% respectively). See p.13 line 330.

3. Page 14 Line 335: "improving bone quality" needs to be rephrased as SOS is not bone quality. I suggest a clear definition of which aspects of bone quality tibia SOS is associated with and this definition is then referred to throughout the manuscript

Thanks for this important comment. We have added definition of bone quality as being an index integrating bone density and cortical architecture, consistent with your above comment 2. Please see p.14 line 336

4. Page 25 Section speed of sound of the tibia and radius: To shorten the manuscript, while still maintaining essential details, I suggest moving the majority of detailed description (also what has been provided in the rebuttal) to supplementary materials, while including the information about reproducibility to the manuscript text body. Given the small observed changes in relation to the respective CVs, this information must not be omitted.

The section has been accordingly changed and shortened and coefficient of variation included.
See p. 25 line 612

Typos:

P395: should read "However, ..."

P14I339: Sos should be SOS

P39 I965 and I969: "is be included in order to" needs to be rephrased

All Typos have been corrected

Reviewer #2 (Remarks to the Author):

The authors have addressed all my comments/concerns.

Many thanks

Reviewer #3 (Remarks to the Author):

The authors have responded satisfactorily to my comments.

Many thanks